# Enhanced flux potential analysis links changes in enzyme expression to metabolic flux

Xuhang Li, Albertha J M Walhout ✉ & L Safak Yilmaz ✉

## Abstract

**Algorithms that constrain metabolic network models with enzyme levels to predict metabolic activity assume that changes in enzyme levels are indicative of flux variations. However, metabolic flux can also be regulated by other mechanisms such as allostery and mass action. To systematically explore the relationship between fluctuations in enzyme expression and flux, we combine available yeast proteomic and fluxomic data to reveal that flux changes can be best predicted from changes in enzyme levels of pathways, rather than the whole network or only cognate reactions. We implement this principle in an 'enhanced flux potential analysis' (eFPA) algorithm that integrates enzyme expression data with metabolic network architecture to predict relative flux levels of reactions including those regulated by other mechanisms. Applied to human data, eFPA consistently predicts tissue metabolic function using either proteomic or transcriptomic data. Additionally, eFPA efficiently handles data sparsity and noisiness, generating robust flux predictions with single-cell gene expression data. Our approach outperforms alternatives by striking an optimal balance, evaluating enzyme expression at pathway level, rather than either single-reaction or whole-network levels.**

**Keywords** Enzyme Expression; Metabolic Network Model; Metabolic Flux; Flux Potential Analysis; Single-cell Data
**Subject Categories** Computational Biology; Metabolism

## Introduction

Metabolism is highly dynamic, with metabolic flux, the rate of metabolic conversion, changing in response to nutritional, environmental, and pathogenic perturbations. Flux can be regulated by multiple mechanisms: metabolite concentrations can directly affect reaction rates; allosteric regulators and covalent modifications can affect the activity of metabolic enzymes; and metabolic enzyme levels can be regulated by changes in gene or protein expression (Fell, 1997). It is commonly postulated that active regulation of enzyme expression levels directly links to flux changes through the corresponding enzyme, i.e., an increase in enzyme level associates with an increase in flux of its associated reactions. This assumption facilitated the generation of hypotheses regarding flux status, leading to many discoveries ranging from tissue-specific metabolism to new disease therapies (Desvergne et al, 2006; Kochanowski et al, 2013; Li et al, 2022; Yizhak et al, 2015). However, studies that simultaneously measure both flux and enzyme levels typically indicate that flux is predominantly regulated by metabolite concentrations rather than enzyme levels, suggesting a weak correlation between flux and the expression of corresponding enzymes (Chubukov et al, 2013; Daran-Lapujade et al, 2004; Daran-Lapujade et al, 2007; Hackett et al, 2016; Lahtvee et al, 2017; Yu et al, 2020). This discrepancy suggests that the relationship between changes in enzyme expression and flux is not well defined, complicating the interpretation of expression data.

A potential gap in studies comparing measured fluxes to enzyme levels is their frequent focus on individual reactions. However, metabolic flux is influenced not only by the enzymes and metabolites directly involved in the reaction of interest (ROI) but also by other reactions in the metabolic network, due to the coupling of fluxes mediated by mass balance at steady state (Kacser and Burns, 1973). Indeed, a plethora of methods have been developed that infer metabolic flux from network-level integration of gene expression data, which were believed to better predict flux (Machado and Herrgard, 2014; Opdam et al, 2017). These methods combine expression data of different metabolic genes based on their connections in the metabolic network to collectively infer flux status. Most past studies focused on either reconstructing context-specific metabolic networks to predict metabolism in tissues and cells (Agren et al, 2012; Becker and Palsson, 2008; Jensen and Papin, 2011; Jerby et al, 2010; Li et al, 2022; Vlassis et al, 2014), or on modeling absolute flux levels with flux balance analysis (FBA), using enzyme levels as additional constraints (Colijn et al, 2009; Lee et al, 2012; O'Brien et al, 2013; Salvy and Hatzimanikatis, 2020; Sanchez et al, 2017). However, these analyses often result in qualitative flux predictions and/or show limited improvement in predictive power over traditional non-integrative FBA approaches in benchmark studies (Machado and Herrgard, 2014; Opdam et al, 2017). More recently, several studies have turned their focus to relative changes in enzyme levels and flux by integrating expression data across different conditions (Kim and Reed, 2012; Pandey et al, 2019; Pusa et al, 2020; Ravi and Gunawan, 2021; Zhu et al, 2017). However, a comprehensive systems-level comparison of experimental flux levels with those predicted by integrating enzyme expression across the metabolic network remains to be done. This

Department of Systems Biology, University of Massachusetts Chan Medical School, Worcester, MA, USA. ✉E-mail: marian.walhout@umassmed.edu; lutfussafak.yilmaz@umassmed.edu

analysis requires a dataset encompassing network-wide measurements of both enzyme and flux levels, coupled with a flexible method capable of mapping and analyzing these relationships at scales ranging from individual reactions to pathways and the entire network.

We recently developed flux potential analysis (FPA), an algorithm that predicts flux changes by integrating relative enzyme levels not only of the enzyme catalyzing the ROI but also the levels of enzymes of nearby reactions (Yilmaz et al, 2020). A critical component of FPA is a distance factor that controls the effective size of the network neighborhood considered, assuming that more distant reactions exert less influence on the flux of the ROI. More recently, a similar algorithm called Compass was developed to analyze single-cell RNA-seq data, producing valuable predictions about metabolic switches in immune cell functions (Wagner et al, 2021). While Compass is mathematically similar to FPA, it combines gene expression data across the entire network without considering the proximity of reactions to the ROI. Given the flexibility of FPA to adjust the integration distance from the ROI, it offers a comprehensive tool that not only covers the capabilities of Compass but also provides adaptability for systematic investigations into how enzyme levels affect flux. However, in the absence of actual flux data, the algorithmic rules and parameters of FPA could not yet be optimized (Yilmaz et al, 2020).

In this study, we systematically explored the associations between enzyme level variations and metabolic reaction flux using published yeast data (Hackett et al, 2016). Initially, we validated that flux correlates more strongly with overall enzyme expression along pathways than with individual reactions. Subsequently, we enhanced the FPA algorithm to more accurately capture the expression data for each ROI and its neighboring reactions, and optimized the distance parameters that govern the pathway length over which expression data is integrated. These improvements resulted in the development of enhanced FPA (eFPA), which surpassed existing methods in predicting relative flux levels from enzyme expression. Applying eFPA to both transcriptomic and proteomic data from human tissues produced similar results, demonstrating eFPA's reliability and its effectiveness in integrating both protein and mRNA levels. Finally, we showcase the applicability of eFPA to single-cell analysis using human single-cell RNA-seq data. Our work establishes that focusing on nearby pathways yields an optimal balance between solely evaluating the genes associated with a ROI and broader network integration, thereby enhancing predictive power. This approach paves the way for a more accurate interpretation of changes in metabolic gene expression in transcriptomic or proteomic data and a better understanding of how such regulation is linked to downstream changes in flux.

# Results

To systematically evaluate the relationship between enzyme expression and flux, we required a dataset that (i) has both flux and enzyme expression data acquired from the same samples; (ii) provides accurate flux values spread across a metabolic network, i.e., not just confined to core carbon metabolism; and (iii) involves multiple conditions to ensure a statistically meaningful analysis. It is difficult to obtain data that covers a broad range of reactions as

traditional methods, such as determining flux through complete isotope balancing, are typically limited to core carbon metabolic pathways (Gopalakrishnan and Maranas, 2015; Kharchenko et al, 2005; Zamboni et al, 2009). Alternatively, network-level flux distributions can be predicted using Flux Balance Analysis (FBA), which incorporates various measured rates as constraints (Antoniewicz, 2015; Hackett et al, 2016). However, the capability to determine flux is frequently restricted by the comprehensiveness of the measured rates. Consequently, only a few studies to date have achieved nearly comprehensive network-level flux determination (Hackett et al, 2016; Kochanowski et al, 2021; Lahtvee et al, 2017). Our literature survey led us to conclude that only a yeast dataset (Hackett et al, 2016) fulfilled the three requirements (Table EV1). In this dataset, flux estimates are available for 232 metabolic reactions, with 156 associated measurements of enzyme levels, across 25 conditions in which different nutrients (glucose, leucine, uracil, phosphate, nitrogen) were limited and the growth rate was titrated (Datasets EV1-EV3).

In order to compare enzyme expression with flux, it is also essential that both variables—enzyme levels and flux—are contextualized appropriately. In yeast cultures, both protein abundance and flux have been shown to scale with specific growth rate (Hackett et al, 2016; Xia et al, 2022), which confounds the interpretation of a direct comparison. In Hackett et al, the proteomic data reflects enzyme abundance in terms of its proportion to the total protein (Hackett et al, 2016), making it intrinsically adjusted for growth variations. To ensure a meaningful comparison with flux, we adjusted the flux data by dividing flux values with the corresponding growth rates. These adjusted, relative flux values are used throughout this study (referred to as flux, see Appendix Text S1 for details).

## Flux changes are associated with pathway-level changes in enzyme levels

To investigate the relationship between enzyme expression and flux, we first analyzed individual reactions and identified those for which flux correlates with the level of the corresponding enzyme(s) (Fig. 1A). A total of 46 of the 156 reactions (29%) showed a significant positive correlation between flux and enzyme levels (false discovery rate (FDR) < 0.05). Hereafter these reactions are referred to as *correlated reactions* (Fig. 1B,C; Appendix Fig. S1 and Dataset EV1). Similar to observations in bacteria (Chubukov et al, 2013), most central carbon reactions in yeast are not correlated, while correlated reactions mainly occur in amino acid and nucleic acid metabolism (Figs. 1D and EV1A–D). These findings agree with a recent study on proteome allocations which suggested that enzyme usage, which indicates how much of an enzyme's available capacity is actually being used in metabolic processes, significantly impacts flux in amino acid biosynthesis (Xia et al, 2022). Many reaction fluxes in arginine, phenylalanine, tryptophan and tyrosine biosynthesis are correlated, which agrees with previous observations (Fig. 1D) (Lahtvee et al, 2017; Moxley et al, 2009). Together, our correlation analysis affirms previously known association between flux and enzyme levels based on a single but comprehensive dataset.

Enzymes that catalyze reactions in the same pathway are often coordinately activated or repressed, thereby enabling network rewiring (Bulcha et al, 2019; Giese et al, 2020; Kochanowski et al,

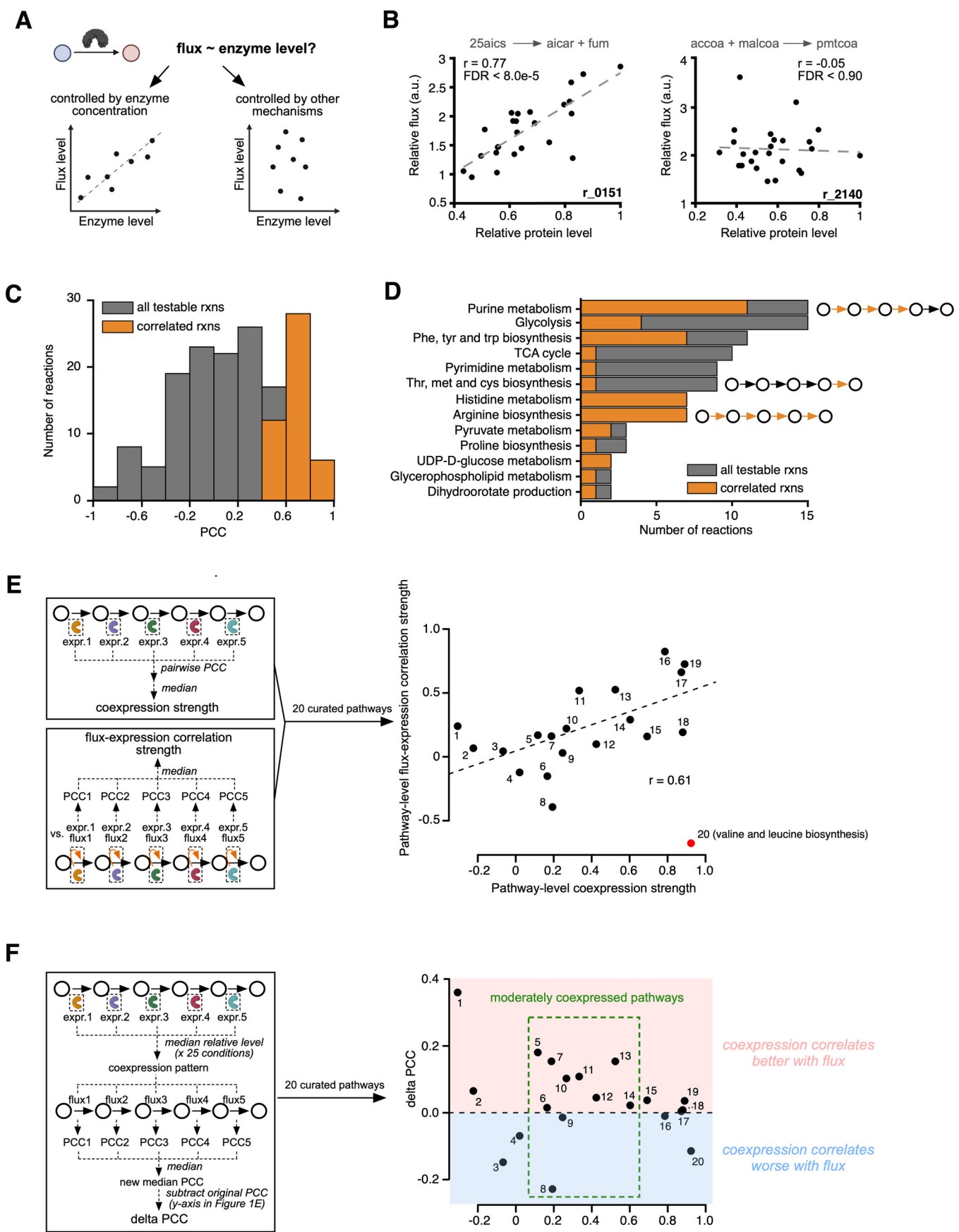

**Figure 1. Pathway-level coordinated regulation of enzyme levels associates with changes in metabolic flux.**

(A) Models for quantitative relations between enzyme levels and flux. (B) Representative examples for flux-enzyme level correlation. Each data point represents a measurement (25 total) of relative flux and enzyme levels for the indicated reaction. The metabolite abbreviations are defined by BiGG model nomenclature(King et al, 2016). a.u., arbitrary units. FDR is defined by the Benjamini-Hochberg adjusted $p$-values from two-tailed test. (C) Pearson Correlation Coefficient (PCC) distribution for the 156 reactions for which both flux and enzyme levels were available. Reactions with significant positive correlation (FDR < 0.05, PCC > 0) are indicated in orange. (D) Metabolic pathway annotations for the 156 reactions. (E) Correlation between pathway-level enzyme coexpression and flux-expression concordance. Pathway information corresponding to the number indices is provided in Fig. EV1E. The trend line was fitted without the outlier (red dot). The correlation coefficient (r) of the trend line is shown in the plot. (F) Comparing flux concordance with individual and pathway-level enzyme expression. Source data are available online for this figure.

2021; Nanda et al, 2023; Watson et al, 2016). Therefore, we next asked whether pathways with a greater degree of enzyme coexpression exhibit an overall better correlation between enzyme levels and flux. We manually defined 20 metabolic pathways that consist of at least three connected reactions for which flux estimations were available, and for which the enzyme levels corresponding to at least two reactions were available (Fig. EV1E and Dataset EV1). Except for valine and leucine biosynthesis, there was a significant correlation between the strength of pathway-level enzyme coexpression and the strength of flux-expression correlation (r = 0.61, $p < 0.006$) (Fig. 1E). The exception of leucine biosynthesis pathway is likely due to the use of a leucine auxotrophic strain (MATa $leu2\Delta1$) (Hackett et al, 2016). This correlation indicates that the more enzymes in a pathway covary in expression under different conditions, the more the flux through the entire pathway correlates with changes in enzyme levels. Based on this observation, we next asked if pathway-level coexpression better predicts flux variations compared to changes in individual enzymes. We defined pathway-level coexpression as a vector of 25 elements that correspond to the 25 conditions, with each element containing the median of relative enzyme levels of all enzymes in the pathway in that condition. Indeed, we found that enzyme coexpression correlates better with pathway flux than the enzyme levels for individual reactions, particularly for pathways that are moderately coexpressed (Fig. 1F). Therefore, we conclude that pathway-level regulation of enzyme expression better predicts flux changes compared with changes in enzyme levels of individual reactions.

## An enhanced flux potential analysis algorithm that translates relative enzyme expression to relative flux

The observation of pathway-level association between flux and enzyme levels suggests that flux predictions may be more effective when focused on pathways in the network neighborhood of a ROI. We reasoned that the most relevant pathways in this neighborhood are likely to extend along reactions that have a linear connection to the ROI, where fluxes are strictly coupled. The linear connections are reflected in the network architecture and can be quantified by the network connectivity degree of metabolites (i.e., the total number of sources and sinks for the metabolite, Fig. 2A). A degree of two indicates strict linearity, and low degrees may generally suggest higher likelihood of flux coupling. To test whether metabolite degree is indicative of flux-expression association, we inspected the 'pairwise cross-informing rate', which defines how likely the flux or expression of one reaction is correlated with the flux or expression of a connected reaction (FDR < 0.05), given the degree of the bridging metabolite that links the two reactions. Indeed, reactions connected by metabolites with a low degree are

more likely to cross-inform each other for both flux and expression as well as expression-flux correlation (Fig. 2B). Although the clarity of this association diminishes with higher degrees, i.e., becoming notably noisier beyond a degree of about 8 (Fig. 2B), these findings support the use of metabolite degree as a heuristic approach to define pathways related to each ROI. Such heuristics can then be incorporated into our FPA algorithm for enhancing network integration.

FPA calculates the relative flux potential (rFP) of an ROI based on relative enzyme mRNA levels across conditions (Yilmaz et al, 2020) (Appendix Text S2). FPA is formulated as an FBA problem whose objective function is the maximal flux of an ROI and whose maximized objective value is named *flux potential*. The key concept of FPA is to convert relative enzyme levels into weight coefficients such that the flux through a reaction is penalized when corresponding enzyme levels are low (Fig. EV2A, Methods) (Yilmaz et al, 2020). The resulting flux potential value is further normalized by its theoretical maximum, i.e., the flux potential in a hypothetical condition that expresses all enzymes at the highest observed level, to obtain a dimensionless rFP that ranges between 0 and 1. FPA uses a distance decay function with which the influence of enzyme levels decreases with distance from the ROI. To fine-tune FPA, we incorporated metabolite degree to utilize flux routes that are both flux-balanced and low in metabolite degree, which can be conceptualized as the related pathways of the ROI (Appendix Text S1) (Fig. EV2B). In addition, we formulated a new distance decay function with a *distance boundary* parameter to precisely control the maximum length of the related pathways where the expression data is integrated, which we hereafter call *integrated pathways* (Figs. 2C and EV2C, Appendix Text S1). We refer to this algorithm as enhanced FPA (eFPA).

Using eFPA, we dissected the association between changes in flux and enzyme expression at various scales of integrated pathways (Fig. 2C). We set each of the 232 reactions for which flux was measured (Hackett et al, 2016) as ROI, and titrated distance boundary from zero to the maximum reaction distance in the network, thus varying pathway definition from the sole ROI to the entire network. For most reactions, we found that the correlation between rFP and flux varied with distance boundary (Figs. 2D and EV3). We refer to the boundary where the correlation is maximal as the *optimal distance boundary* and to the corresponding predictions as the results of *optimal-boundary eFPA*. Overall, the flux of 44% (101/232) of the reactions could be significantly correlated with rFP (Appendix Fig. S2A). Importantly, with an optimal boundary, eFPA improved the correlation for most reactions, further confirming that flux changes are associated with enzyme expression in related pathways (Fig. 2E). Most reactions showed high correlation when the distance boundary was set between 2 and 6 (Fig. EV3), and the total

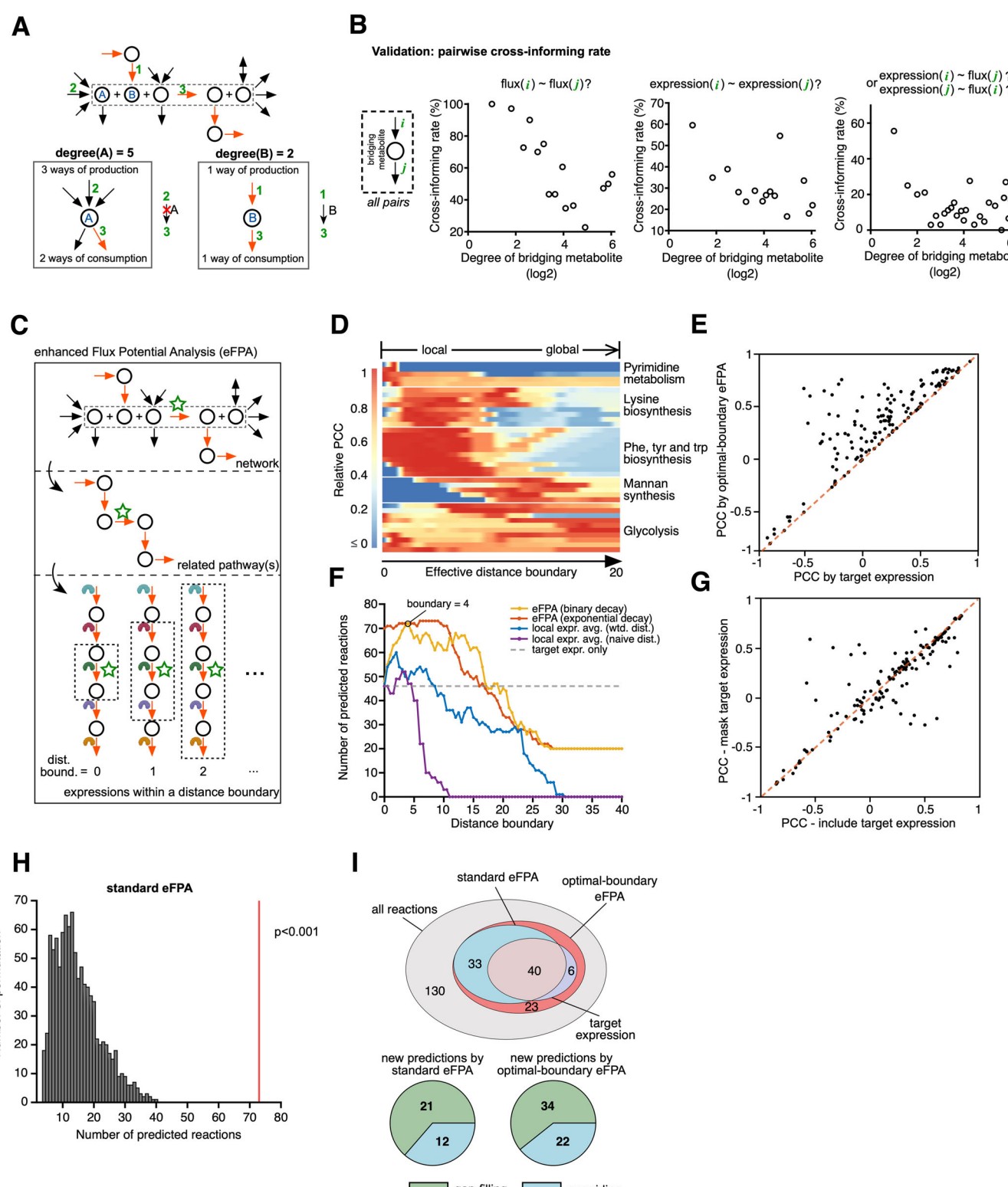

number of significantly correlated reactions peaked at a distance boundary of 4 (Fig. 2F, yellow curve). As expected, the correlation with flux diminished when gene expression was solely considered for the ROI (Fig. 2F, blue and purple curve). These observations together indicate that where flux and

expression levels are associated, this relationship is mostly localized to short pathways.

Ideally, different distance boundaries need to be considered for each ROI. However, this is not practical. Therefore, we developed a robust variant of eFPA that is not sensitive to the choice of distance

**Figure 2. Enhanced flux potential analysis (eFPA) predicts flux changes by leveraging pathway-level enzyme expression.**

(A) Toy network illustrating the degree of bridging metabolites. Circles are metabolites and arrows are reactions. The red arrows indicate a hypothetical pathway for the ROI inside the dotted box. (B) Analysis of pairwise cross-informing rate. (C) Schematic for eFPA. The ROI is labeled by a green star. (D) Representative results of eFPA for different reactions. Values are row-normalized flux-rFP PCCs. Rows correspond to different ROIs and columns to the effective distance boundary that indicates the maximum length of pathway(s) integrated (see Methods). (E) Comparison of the correlation of flux with ROI rFP and ROI expression. (F) Total number of predicted reactions with different approaches. Local expression average was defined by the average expression of reactions that are connected to ROI within a distance boundary measured by naïve or weighted distance. (G) Comparison of standard eFPA predictions with and without inclusion of enzyme levels corresponding to the ROI. Values are flux-rFP PCCs. (H) Randomization test of the standard eFPA. The histograms show the distribution of the number of reactions predicted in 1000 permutations and the red line indicates the real observation. The empirical p-value from this permutation test is labeled by the red line. (I) Classification of fluxes predicted by different types of eFPA. Source data are available online for this figure.

boundary for different reactions, and thus can be broadly used for different expression datasets. We optimized the distance decay function of eFPA and found that an exponential decay (Fig. EV2C) helped to predict relative flux for a broader range of distance boundaries (Fig. 2F, red curve). We refer to the use of exponential decay function with a distance boundary of 6 as "standard eFPA", which is independent of the fitted optimal distance boundaries. To test the predictive power of this tool more directly, we asked if robust flux predictions can be obtained even when enzyme levels of the ROI were ignored. Indeed, only using levels of enzymes catalyzing reactions other than the ROI predicted reaction fluxes equally well as when ROI's enzyme levels were included (Fig. 2G). In addition, we evaluated the significance of eFPA predictions by a randomization that shuffled the reaction labels in the input expression data. This analysis clearly showed that pathway-level flux-expression association is an intrinsic property of the flux-expression relationship and not a result of overfitting (Fig. 2H; Appendix Fig. S2B). Overall, eFPA correctly predicted the relative flux of a large proportion of metabolic reactions, including ROIs for which enzyme level measurements were not available (*gap-filling predictions*) and ROIs for which their own enzyme levels did not correlate with flux (*overriding predictions*) (Fig. 2I). The remaining 130 reactions likely represent cases where fluxes are not controlled or correlated with enzyme expression. Instead, these fluxes may be driven by metabolite levels or regulated through mechanisms such as allostery, which are not directly influenced by changes in enzyme expression.

We evaluated the predictive power of standard eFPA through a benchmark analysis against alternative methods (Fig. 3A). Standard eFPA outperformed three recent algorithms, ΔFBA (Ravi and Gunawan, 2021), Compass (Wagner et al, 2021) and REMI (Pandey et al, 2019), all of which predict flux changes based on variations in enzyme expression. A critical factor in this performance increase is the selective integration of ROI-proximate network neighborhoods: removing the distance decay from FPA diminished its power, whereas incorporating a decay function into Compass boosted its predictivity (Fig. 3A). In addition, adjusting the distance order parameter in FPA, corresponding to the distance boundary in eFPA (Fig. EV2C), was insufficient to match the predictive power achieved by eFPA (Fig. 3B), further emphasizing the significance of the enhancements in our approach (Yilmaz et al, 2020). This difference is largely driven by the weighted distance (Fig. 3B), which incorporates the metabolite degree heuristics (Fig. EV2B). For instance, FPA predictions showed a poor correlation with flux changes for r_0939 in tyrosine biosynthesis, a reaction without available enzyme level data, whereas eFPA achieved significantly improved accuracy (Fig. 3C). Analysis of the integration process revealed that FPA predictions were influenced by protein

expressions from glycolysis (e.g., r_0892) and the electron transport chain (r_0770), which were unrelated to the flux of the ROI (r_0939). These reactions appeared proximal in the network under the standard distance metric, compromising the original FPA predictions. However, eFPA effectively corrected for this issue by using weighted distance, which assigned larger values to these reactions due to high-degree metabolites (e.g., pyruvate) in the pathways connecting them to the ROI. Consistent with these observations, incorporating a weighted distance into FPA nearly restored the correlation, demonstrating its critical role in improving predictive accuracy (Fig. 3C).

To further illustrate how eFPA predicts flux, we used the proline synthesis pathway as an example (Fig. 3D). This pathway includes four reactions with only the first reaction, r_0468, showing statistically significant flux-enzyme correlation (Fig. 3D). However, eFPA predictions significantly correlated with relative fluxes in all reactions of this pathway, including r_1887, which lacks associated enzymes and therefore can only be predicted by data integration along the pathway. Although the correlation between flux and rFP values was weakened by conditions with inconsistent expression levels and flux (e.g., red data points, Fig. 3D), it remained significant due to conditions with consistently high expression levels corresponding to high flux (dark blue data points, Fig. 3D). Notably, the enzyme levels for the last pathway reaction, r_0957, showed minimal variation across the 25 conditions and no meaningful correlation with flux (Fig. 3D), suggesting that this reaction is not regulated by enzyme expression. Thus, eFPA can not only predict flux for reactions lacking associated enzymes but also for those not regulated by their enzyme levels. Overall, eFPA demonstrated optimal performance for this pathway: while reaction expression gave the best correlation for a single reaction, eFPA achieved significant correlations across all reactions despite compromises from integrating data from poorly correlated reactions. The original FPA algorithm also captured some flux-expression relationships but performed suboptimally (Fig. 3D). Taken together, these results suggest that metabolic flux changes correlate more efficiently with coordinated enzyme expression changes in pathways, a principle computationally best realized in the eFPA algorithm.

## Analyzing human tissue metabolism by eFPA

Next, we asked whether eFPA can be used to study human metabolism, specifically at the level of different tissues. We performed standard eFPA on a human dataset containing protein and mRNA levels for 32 tissues (Jiang et al, 2020) and the human metabolic network model, Human 1 (Robinson et al, 2020) (Fig. 4A). To interpret the predicted rFPs for tissue enrichment,

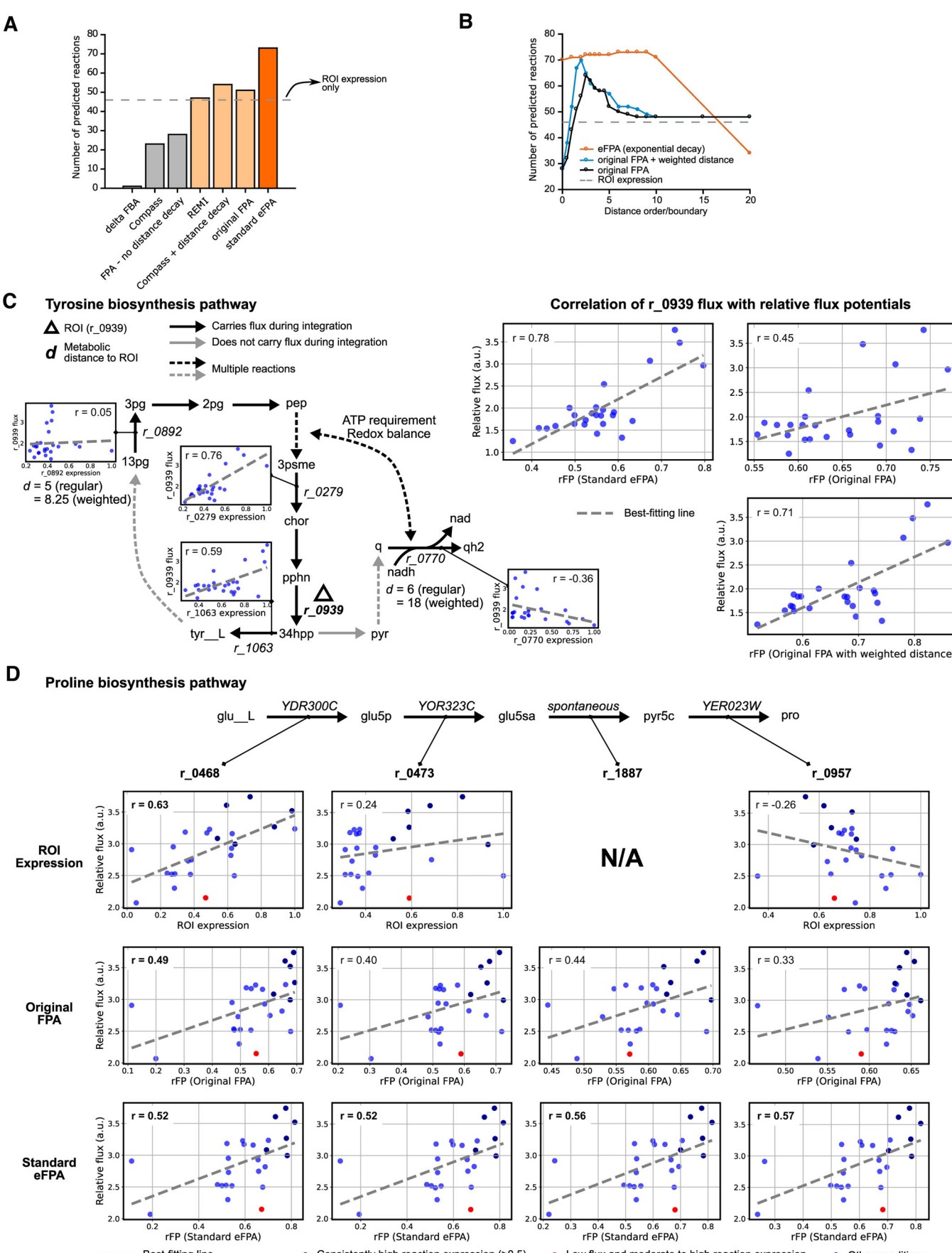

**A**

**B**

**C** Tyrosine biosynthesis pathway

Correlation of r_0939 flux with relative flux potentials

**D** Proline biosynthesis pathway

**Figure 3.  Benchmark analysis of eFPA.**

(A) The number of reactions predicted using standard eFPA and alternative methods. Predicted reactions are defined as reactions whose measured fluxes across the 25 conditions significantly (FDR < 0.05, PCC > 0) correlate with the predicted relative flux levels using expression data. Gray, yellow, and orange bars indicate methods whose predictive power were below, around, and above the power using solely enzyme changes of the ROIs (dashed line), respectively. The 'FPA – no distance decay' group refers to the predictions by the original FPA with a distance order parameter of 0, which disables the distance decay function. The 'Compass + distance decay' group refers to the predictions by a modified Compass algorithm that integrates a distance decay function from the original FPA with a distance order parameter of 2.5. The configurations of each algorithm are detailed in Appendix. (B) Total predicted reactions by FPA and eFPA as a function of their respective metabolic distance parameters: distance order for FPA and distance boundary for eFPA. Original FPA with weighted distance algorithm uses the weighted distance metric to define metabolic distances, similar to eFPA, but retains the original FPA distance decay function (as shown in Fig. EV2C). (C) Example ROI r_0939, whose successful flux prediction requires the use of weighted distance. The pathway containing this reaction (left) and predictions with eFPA and FPA (right) are shown. No reaction expression data is available for this ROI. The correlation between the relative flux of r_0939 and the reaction expression of four integrated reactions is shown on the pathway. The distances to two reactions with poor correlation are indicated using both regular and weighted distance metrics. Shortest pathways between the ROI and these reactions, used to calculate these distances, are highlighted. While these pathways do not contribute to data integration in this case, they define the shortest distances to the ROI. (D) Proline synthesis pathway as an example illustrating the optimality of eFPA predictions compared to reaction expression and original FPA. Data points representing conditions with a clear and observable influence on the correlations are highlighted. PCC in bold indicates statistically significant correlations (FDR < 0.05). Source data are available online for this figure.

we defined a tissue enrichment score as the difference of an rFP value from the median rFP of 32 tissues ($\Delta rFP$). Using these scores, we generated a comprehensive landscape of predicted human tissue-enriched metabolic fluxes using either protein or mRNA levels as input (Fig. 4B; Appendix Fig. S3, Dataset EV4). Importantly, flux predictions based on protein or mRNA data are highly consistent across tissues. Therefore, mRNA levels can effectively be used in eFPA.

The predicted flux landscape revealed diverse metabolic specializations of various tissues (Fig. 4B; Appendix Fig. S3). Many of these are consistent with existing knowledge, such as bile acid biosynthesis in liver (Russell, 2003), active metabolism of membrane lipids and signaling molecules in brain (Barber and Raben, 2019; Hanley et al, 1988), and energy metabolism in muscle (Westerblad et al, 2010). Our predictions further yielded many hypotheses for future studies, for instance, a high metabolic similarity between lung and spleen that was driven by reactions involved in immunometabolism (e.g., synthesis of leukotrienes) (Fig. 4B). Finally, some predictions were strictly derived by standard eFPA. For example, peroxisomal oxidation of palmitoyl-CoA was predicted to be enriched in heart, although this organ was not enriched for the expression of associated enzymes (Fig. 4C; Appendix Fig. S4). Interestingly, this prediction is consistent with the stable isotope tracing data in mice where heart was found to be the top organ that uses fatty acids as a TCA carbon source (Hui et al, 2020) (Fig. 4C). Altogether, these results indicate that eFPA can derive meaningful predictions that are enabled by the pathway-level integration of enzyme expression.

## eFPA enables metabolic analysis at single-cell level

Single-cell RNA-seq (scRNA-seq) is an emerging method that delineates the transcriptomes of individual cells (Kolodziejczyk et al, 2015). However, this method often has low sensitivity, which can result in numerous zero counts of gene expression. These zero counts are frequently caused by dropout events, where an mRNA is present in the cell but fails to be detected due to technical limitations, rather than its true absence (Luecken and Theis, 2019). Since standard eFPA integrates not only the expression of genes associated with a ROI, but also the expression levels from enzymes catalyzing surrounding reactions, we wondered whether it could significantly enhance the metabolic analysis of scRNA-seq data beyond what was achievable with previous approaches. We

extracted a diverse dataset from the Tabula Sapiens database (Tabula Sapiens et al, 2022) that includes 2264 cells across 23 cell types from 13 organs, and integrated this data with the Human 1 model using each of the 3002 cytosolic reactions as ROI (Fig. 5A). We compared the results with two other metrics: solely the expression levels of enzymes associated with each ROI, and Compass (Wagner et al, 2021), which integrates network reactions but does not incorporate a distance decay. The original Compass method offers a partial lumping of multiple cells (i.e., information sharing) to mitigate the sensitivity issue. To test the integrations at the actual single-cell level, we skipped this approach in our implementation and named it Compass-.

We hypothesized that eFPA achieves an optimal integration balance between reaction expression and Compass-. Reaction expression has limited analytical depth as it does not consider information beyond the ROI. In contrast, Compass- does incorporate network-wide gene expression information, but this can potentially dilute local signals by over-emphasizing distant gene expression. Given that eFPA is designed to selectively integrate local, pathway-level gene expression data, we reasoned that it would be the most effective approach for generating biologically meaningful predictions. Initial tests using Principal Components Analysis (PCA) supported this: eFPA predictions showed better clustering and retained higher explained variance in the top principal components compared to ROI expression alone (Fig. 5B,C). Although Compass- explained an even broader variance due to its comprehensive network approach, it tended to blur the distinctions among cell types in the PCA plot (Fig. 5B, e.g., orange cells became less distinguishable from blue cells). Thus, eFPA strikes a balance, capturing enough variance for clear clustering without over-generalizing, which can obscure finer distinctions among cell types.

We next counted the number of cell-type-enriched metabolic functions predicted by the three approaches. We conservatively defined such predictions as reaction fluxes that are significantly higher in a particular cell type compared to the main population, with a Fold Change (FC) greater than 1.2 and a p-value less than 1E−10 (Fig. 5D). This approach resulted in numerous reactions that are enriched in many cell types, with liver hepatocytes, heart muscle cells, and bone marrow erythroid progenitors having the highest predicted metabolic activity (Fig. 5E). In all cell types, eFPA, either alone or in agreement with one or both of the other two methods, covered the majority of predictions. Reaction expression alone also

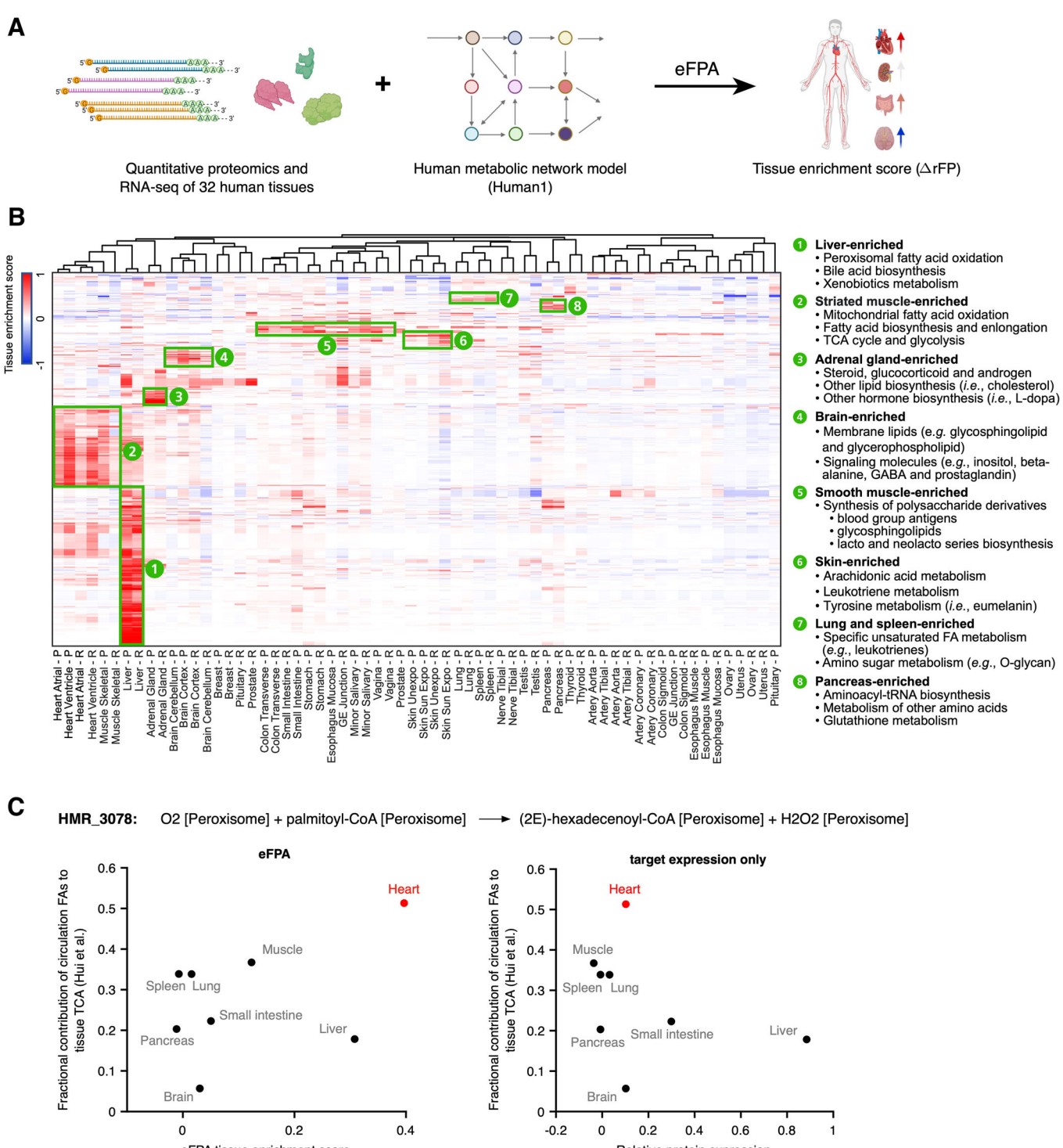

**Figure 4. eFPA predictions for human tissue metabolism.**

(A) Cartoon of eFPA analysis. rFP values were centered by subtracting the median rFP of a reaction across all 32 tissues, and the difference from median (△rFP) was referred to as the tissue enrichment score. (B) Heatmap of the △rFP of 3441 tissue-enriched reactions in 32 human tissues. Tissue-enriched reactions were defined with △rFP greater or equal to 0.2 in at least one tissue based on either protein or mRNA expression data as the input. Only high-confidence predictions are shown (see Methods). Column label shows tissue names with a single letter suffix indicating the prediction was made from protein (-P) or mRNA (-R) data. (C) Predictions of fatty acid oxidation flux in tissues based on eFPA (left) or expression levels (right) of the ROI (formula indicated at the top). Protein levels were used in this analysis. Y-axis indicates experimental data of the tissue TCA contributions from circulation fatty acids (oleic acid, linoleic acid, and palmitic acid together) in mice (Hui et al, 2020) (see Appendix for details). Source data are available online for this figure.

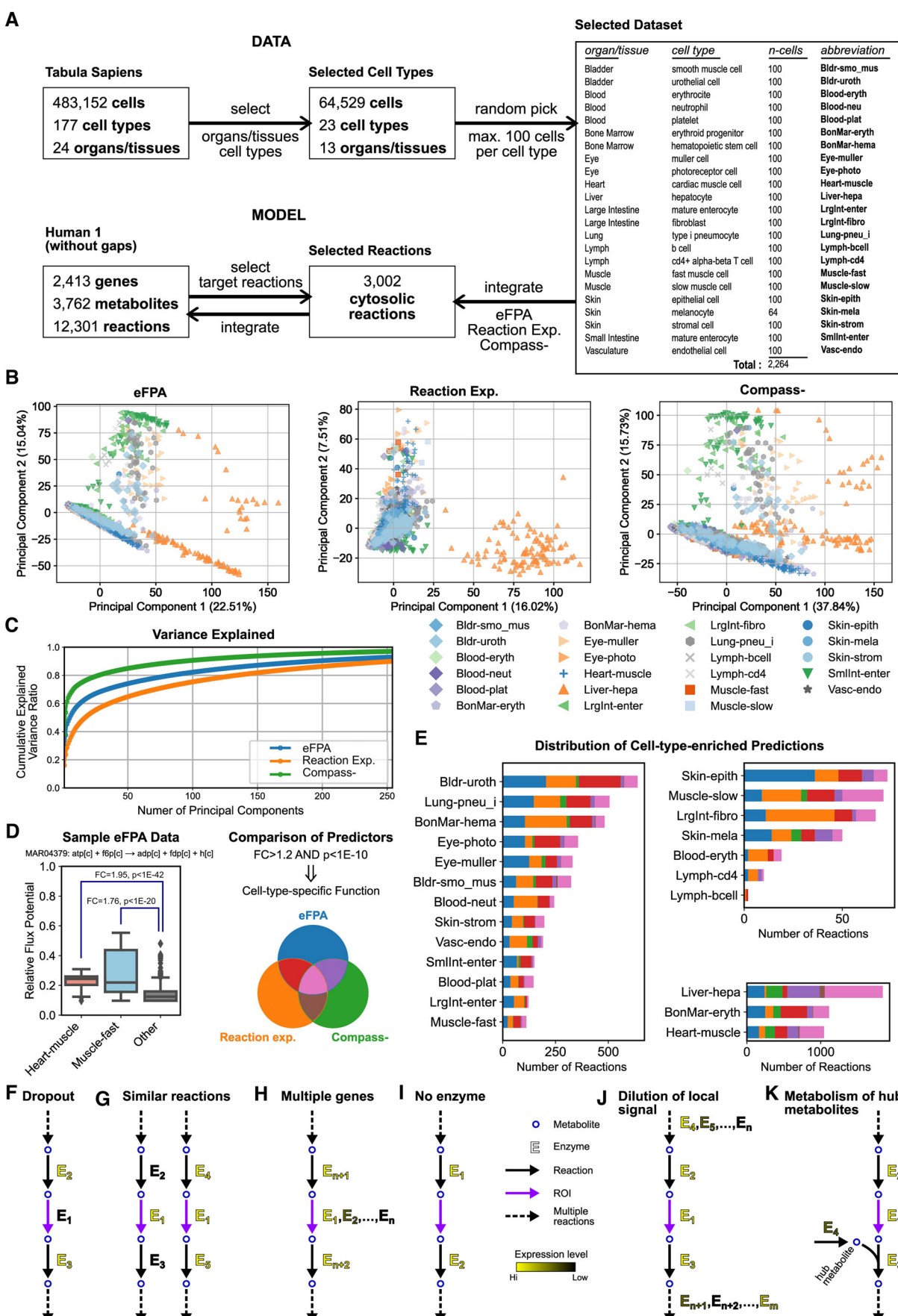

◀ **Figure 5. Integration of single-cell data.**

(A) Overview of the dataset, model, and integration methods used. Abbreviations for cell types, shown in the right panel, are referenced in subsequent figures and Table 1. (B) Principal Component Analysis (PCA) of predicted relative reaction flux potentials using different methods. The variance explained by each principal component is indicated on the x- and y-axis labels as percentages. (C) Cumulative variance explained by principal components, ranked according to their importance. The analysis includes enough principal components to account for at least 90% of the variance explained by any method. (D) Definition of cell-type enrichment used in the analysis. The example box plot illustrates how the glycolysis reaction MAR04379 exhibits relatively high flux potential in heart muscle and fast muscle cells compared to all others. The p-values for these comparisons are 1.8E−43 for heart muscle and 2.3E−21 for fast muscle, respectively. The plots display the median (central line), interquartile range (IQR; box boundaries), and whiskers extending to the nearest data points within 1.5*IQR from the first and third quartiles. Points outside this range are depicted as outliers when present. Distributions are shown for each cell type with the number of data points (n) per cell type derived from the table in (A). Specifically, n values are 100 for both heart muscle and fast muscle, and 2064 for the "Other" category, which includes all remaining data points not covered by the listed cell types. A reaction is considered enriched to a cell type if the fold change (based on the median) and p-value (from a statistical comparison based on Wilcoxon rank-sum test) meet specified criteria when comparing the predicted values for that cell type against all others, except for the highest ranking cell type (or except for the second highest if the compared cell type ranks at the top; see Methods for details). A Venn diagram illustrates the comparison of cell-type-enriched reactions and provides the color scheme used in (E). (E), Distribution of cell-type-enriched reactions for each cell type, with colors indicating the source of enrichment predictions as defined in (D). Three plots group cell types according to the number of enriched reactions. (F–K) Instances where Reaction Expression (F–I) or Compass- (J, K) fail to capture the actual flux potential of ROIs.

produced a significant number of cell-type-enriched predictions, but Compass- alone typically yielded a much smaller number, except for cell types of high general metabolic activity. Importantly, there were only few predictions shared by Compass- and reaction expression but not by eFPA (Fig. 5E, brown).

We reasoned that predictions solely made by reaction expression are less reliable because it means that expression levels from enzymes that catalyze surrounding reactions do not support the ROI expression. In other words, if the expression changes of surrounding reactions were coherent with that of a ROI, the flux potential of that ROI would then also have been captured by eFPA and/or Compass-. Given that eFPA recalled vast majority of other predictions, we next focused on the predictions that were not always in agreement with the other two methods and identified those predictions that could also be verified with existing knowledge (Table 1). We studied these instances in detail to understand why eFPA recalled them as true positives, while one or both of the other methods did not. We determined several expected scenarios that are particularly challenging for reaction expression (Fig. 5F–I) or Compass- (Fig. 5J,K).

For example, reaction expression often fails to detect active ROIs when the associated enzyme appears unexpressed likely due to dropout events (Fig. 5F). This approach also struggles when an enzyme functions in multiple pathways, causing all associated ROIs to seem equally active (Fig. 5G). Complications also arise with multiple isozymes, including paralogs with uncertain annotation and isozymes active in different cell types, which can obscure the true activity signal of the ROI (Fig. 5H). Moreover, reaction expression cannot account for ROIs without associated enzymes, such as non-enzymatic reactions or those catalyzed by unidentified enzymes (Fig. 5I). Both eFPA and Compass- address these issues by integrating gene expression from other network reactions alongside the ROI. However, Compass- might dilute these local signals since it considers the entire network (Fig. 5J). Further, even when few genes are integrated, the involvement of hub metabolites, which participate in numerous processes, can introduce irrelevant gene expression data (Fig. 5K). Several specific examples illustrate how these issues manifest and how eFPA effectively resolves the introduced complexities (Fig. 6).

The first example focuses on two ROIs in glycogen metabolism (Fig. 6A), one producing (MAR05398) and the other converting (MAR01380) phosphorylase-limit dextrin (dxtrn). Both reactions were predicted by eFPA to have high flux potential in all three

muscle cell types, a prediction not as strongly supported by the other two methods (Fig. 6B), but consistent with general knowledge (Cusso et al, 2003; Testoni et al, 2017). To understand the mechanism of prediction, we focused on two fast muscle cells, which have opposite levels of flux potentials for MAR05398 according to eFPA and reaction expression (Fig. 6A,B; the two cells are represented by a square and a circle). For Cell #1 (square), eFPA predicted high flux in MAR05398 even though the reaction expression was low (Fig. 6B). Indeed, we found that this cell exhibits high expression (low penalty) throughout the pathway, except for two reactions including this ROI (Fig. 6A). These two reactions share the same gene-protein-reaction (GPR) association, and we reasoned that the lack of their expression may be due to technical dropouts (Fig. 5F). Thus, the eFPA prediction reflects the integration of the relevant pathway and avoids potential misinterpretation from noise. In contrast, Cell #2 shows low flux potential in MAR05398, but high expression (Fig. 6B, circle). However, this high expression likely reflects the activity of the other reaction that shares the same GPR with MAR05398, but is located in the dxtrn degradation branch of the pathway (Figs. 5G and 6A). Indeed, reactions along the degradation branch are well expressed, while the branch that produces dxtrn is poorly expressed in Cell #2 (Fig. 6A). Consistently, the flux potential of MAR01380 on the degradation branch is predicted to be high by eFPA in this cell (Fig. 6B).

Meanwhile, the potential for both reactions in most skeletal muscle cells was masked in Compass- integration due to strong negative signals (i.e., low expression) from network reactions such as diphosphatase that acts as a source of phosphate (pi) (Figs. 5K and 6A). As with the original FPA (Yilmaz et al, 2020), eFPA does not assume proximity of reactions through side metabolites such as pi and associates reactions connected by these to the ROI with a high distance, thus filtering out their influences. Overall, the eFPA algorithm perfectly captured the human interpretation of glycogen metabolism at the pathway-level for specific ROIs and was the only resource to provide reliable predictions for all muscle cells.

The second example (Fig. 6C) involves two ROIs that produce (MAR02551) and modify (MAR02553) leukotriene B5 (LT-B5). These reactions were consistently, and likely correctly (Strasser et al, 1985; von Schacky et al, 1990), predicted to be most active in neutrophils solely by eFPA (Fig. 6D). The flux potential of MAR02553 was underestimated by both reaction expression and Compass- for distinct reasons. For reaction expression, we found

**Table 1. Examples of cell-type-enriched, verifiable reaction activity predicted by single-cell data integration.**

| Process | Reaction[a] | Cell type | eFPA | | | Reaction expression | | | Compass | | | Fig.[b] | Ref. |
|---|---|---|---|---|---|---|---|---|---|---|---|---|---|
| | | | rank | FC | $-\log_{10}(p)$ | rank | FC | $-\log_{10}(p)$ | rank | FC | $-\log_{10}(p)$ | | |
| Glycogen metabolism | MAR05398 (f) | Heart-muscle | 1 | 2.15 | 44.4 | 3 | 2.60 | 24.3 | 1 | 1.25 | 44.7 | 5A | (Depre et al, 1999; Testoni et al, 2017) |
| | | Muscle-slow | 2 | 1.75 | 21.5 | 2 | 4.33 | 28.9 | 13 | 1.01 | 1.47 | | (Adeva-Andany et al, 2016; Cusso et al, 2003) |
| | | Muscle-fast | 3 | 1.69 | 16.5 | 1 | 4.33 | 48.2 | 9 | 1.06 | 6.56 | | (Adeva-Andany et al, 2016; Testoni et al, 2017) |
| Glycogen metabolism | MAR01380 (f) | Heart-muscle | 1 | 1.83 | 48.4 | 1 | 3.46 | 44.1 | 1 | 1.21 | 45.4 | 5A | (Depre et al, 1999) (Testoni et al, 2017) |
| | | Muscle-fast | 2 | 1.42 | 35.4 | 3 | 1.00 | 5.3 | 9 | 1.05 | 6.9 | | (Adeva-Andany et al, 2016; Testoni et al, 2017) |
| | | Muscle-slow | 4 | 1.32 | 21.2 | 8 | 1.00 | 0.245 | 13 | 1.02 | 2.51 | | (Adeva-Andany et al, 2016; Cusso et al, 2003) |
| Leukotriene B5 metabolism | MAR02551 (f) | Blood-neut | 1 | 1.53 | 35.7 | 5 | 1.00 | 14.5 | 2 | 1.13 | 27.3 | 5B | (Strasser et al, 1985) |
| | MAR02553 (f) | Blood-neut | 1 | 1.77 | 43.9 | 2 | 1.25 | 59.8 | 18 | 0.98 | 5.46 | | (von Schacky et al, 1990) |
| Adenosyl succinate formation from IMP | MAR04042 (f) | Muscle-fast | 1 | 1.32 | 22.8 | 1 | 6.00 | 42.1 | 2 | 1.20 | 18.7 | 5C | (Lowenstein and Goodman, 1978; Tornheim and Lowenstein, 1972) |
| | | Muscle-slow | 2 | 1.31 | 16.4 | 2 | 6.00 | 33 | 8 | 1.08 | 8.51 | | (Lowenstein and Goodman, 1978; Tornheim and Lowenstein, 1972) |
| Adenosyl succinate conversion to AMP | MAR04412 (r) | Muscle-fast | 1 | 1.31 | 22.6 | 3 | 1.61 | 3.55 | 2 | 1.20 | 18.7 | 5C | (Lowenstein and Goodman, 1978; Tornheim and Lowenstein, 1972) |
| | | Muscle-slow | 2 | 1.30 | 16.2 | 15 | 0.76 | 0.846 | 8 | 1.08 | 8.51 | | (Lowenstein and Goodman, 1978; Tornheim and Lowenstein, 1972) |
| Purine metabolism | MAR04812 (r) | BonMar-eryth | 1 | 1.99 | 38.2 | 1 | 1.87 | 21.3 | 1 | 1.29 | 36 | 5C | (Maynard et al, 2024) |
| | | BonMar-hema | 2 | 1.33 | 23.8 | 5 | 1.54 | 11.8 | 5 | 1.09 | 12.9 | | (Karigane et al, 2016) |
| Purine metabolism | MAR04406 (f) | BonMar-eryth | 1 | 1.94 | 38.3 | 1 | 1.01 | 47.3 | 1 | 1.29 | 36 | EV4A | (Maynard et al, 2024) |
| | | BonMar-hema | 2 | 1.35 | 25.3 | 3 | 1.00 | 9.64 | 5 | 1.09 | 12.9 | | (Karigane et al, 2016) |
| Myo-inositol metabolism | MAR06571 (f) | Heart-muscle | 1 | 1.67 | 41 | 5 | 1.70 | 9.76 | 1 | 1.61 | 41 | EV4B | (L'Abbate et al, 2020) |
| Carnitine shuttle | MAR02755 (r) | Liver-hepa | 1 | 1.60 | 36 | 6 | 1.00 | 6.85 | 2 | 1.11 | 18.7 | EV4C | (Longo et al, 2016) |
| Methylglyoxal detox. | MAR03853 (f) | Liver-hepa | 1 | 1.42 | 34.1 | 13 | 1.39 | 0.262 | 1 | 1.30 | 39.4 | EV4D | (Seo et al, 2014) |
| Xenobiotics detox. | MAR07100 (f) | Liver-hepa | 1 | 2.15 | 63.5 | 1 | 1.51 | 23.5 | 1 | 1.47 | 62.3 | EV4E | (van Vugt-Lussenburg et al, 2022) |
| Glycosphingolipid mod. | MAR00891 (f) | Bldr-uroth | 1 | 1.33 | 29.2 | 1 | 1.54 | 23 | 3 | 1.11 | 15.8 | EV4F | (Watanabe et al, 2022) |
| Putrescine degradation | MAR08604 (f) | Bldr-smo_mus | 2 | 1.50 | 40.3 | 1 | 2.32 | 54.2 | 3 | 1.13 | 19 | EV4G | (Nakase et al, 1990) |
| Dopamine prod. | MAR06731 (f) | Eye-photo | 1 | 3.42 | 134 | 1 | 11.00 | 255 | 1 | 1.43 | 47.9 | EV4H | (Zhou et al, 2017) |
| | | Eye-muller | 3 | 1.00 | 32.5 | 2 | 1.00 | 72.1 | 4 | 1.13 | 17 | | (Kubrusly et al, 2008) |
| superoxide dismutase | MAR03960 (f) | Skin-strom | 1 | 1.62 | 49.3 | 1 | 1.72 | 49.4 | 8 | 1.02 | 5.85 | EV4I | (Altobelli et al, 2020) |
| | | LrgInt-fibro | 3 | 1.46 | 30 | 3 | 1.53 | 30 | 11 | 1.01 | 2.84 | | (Arcucci et al, 2011) |
| p-Cresol detox. | MAR10460 (f) | Lung-pneu_i | 1 | 1.90 | 52.7 | 1 | 2.83 | 48.2 | 2 | 1.16 | 42.1 | EV4J | (Chang et al, 2018) |
| | | LrgInt-enter | 2 | 1.47 | 20.1 | 2 | 1.77 | 28.4 | 3 | 1.10 | 22.8 | | (Clayton et al, 2009) |
| Putrescine prod. | MAR04212 (f) | Lymph-bcell | 2 | 1.97 | 11.2 | 2 | 2.54 | 14.2 | 9 | 1.15 | 3.41 | EV4K | (Hesterberg et al, 2018) |
| Eumelanin prod. | MAR08540 (f) | Skin-mela | 1 | 1.63 | 302 | NA | NA | NA | 1 | 1.18 | 23.2 | EV4L | (Slominski et al, 2022) |
| 6-keto-prostaglandin prod. | MAR01320 (r) | Vasc-endo | 2 | 2.49 | 259 | NA | NA | NA | 2 | 1.47 | 36.7 | EV4M | (Patel et al, 1983) |

**Table 1.** (continued)

| Process | Reaction[a] | Cell type | eFPA | | | Reaction expression | | | Compass- | | | Fig.[b] | Ref. |
|---|---|---|---|---|---|---|---|---|---|---|---|---|---|
| | | | rank | FC | $-\log_{10}(p)$ | rank | FC | $-\log_{10}(p)$ | rank | FC | $-\log_{10}(p)$ | | |
| Coproporphyrin I prod. | MAR04773 (r) | BonMar-eryth | 1 | 2.24 | 77.9 | NA | NA | NA | 1 | 1.48 | 51.1 | EV4N | (Besnard et al, 2020) |
| | | BonMar-hema | 3 | 1.27 | 14.5 | NA | NA | NA | 7 | 1.04 | 7.2 | | (Tezcan et al, 1998) |
| Tripeptides metabolism | MAR10981 (f) | SmlInt-enter | 1 | 1.11 | 49.3 | NA | NA | NA | 1 | 1.11 | 7.69 | EV4O | (Miner-Williams et al, 2014) |
| | | LrgInt-enter | 3 | 1.00 | 21 | NA | NA | NA | 18 | 0.97 | 0.152 | | (Madhavan et al, 2011) |

[a]Parenthetical information indicates whether the reaction activity concerns the forward (f) or reverse (r) direction with respect to its formula in the model.
[b]Single-cell data for the row is shown in the figures indicated.

that the prediction was complicated by the involvement of multiple cytochrome p450 genes, each linked to dozens of other reactions across the metabolic network (Fig. 6E). Only one of these genes is significantly expressed in neutrophils (Fig. 6E, CYP4F3). Consequently, this example shows a severe case for both ambiguous GPR with multiple genes (Fig. 5H) and parallel reactions with similar GPR (Fig. 5G), where the interpretation of reaction expression is often confounded and the effective assessment of the likelihood of activity requires contextual analysis within the network, as done by eFPA and Compass-. However, Compass- failed to recognize this activity due to a network constraint that allows the product of MAR02553, 20OH LT-B5, to drain only if various other arachidonates are simultaneously produced, as there is a single reaction which ties them together (Fig. 6C, MAM10003). This requirement creates an extreme case for signal dilution (Fig. 5J) in Compass- integration, which uniformly processes all network reactions.

Both Compass- and reaction expression performed reasonably well in predicting the activity of the other reaction, MAR02551 (Fig. 6C,D), aided by a clear transport system for unmodified LT-B5 and a GPR involving a single gene (LTA4H). Nevertheless, only eFPA consistently ranked neutrophils highest, with a robust and realistic distribution of predicted activities across this cell population (Fig. 6D). Overall, this example highlights eFPA as a robust predictor, demonstrating better performance not only in the presence of GPR ambiguities and network complexities but also in more straightforward scenarios.

The third example examines the two-step conversion of IMP to AMP through adenosyl succinate (dcamp) by the ROIs MAR04042 and MAR04412 (Fig. 6F). This conversion is known to occur in muscle cells as part of the purine nucleotide cycle (Lowenstein and Goodman, 1978), which is clear for the first reaction from reaction expression (Fig. 6G). However, reaction expression failed to recognize the high activity of muscle cells for the second reaction (MAR04412) and Compass- failed for both reactions (Fig. 6G). The prediction of the second reaction is complicated due to the involvement of genes shared with another reaction in purine biosynthesis, MAR04812 (Figs. 5G, 6F,H). These shared genes are more prominently expressed in erythroid progenitors, likely reflecting the significant role of MAR04812 in their purine biosynthesis (Maynard et al, 2024), which was successfully predicted by all three approaches. However, reaction expression inaccurately suggests that erythroid progenitors as the primary site of MAR04412 activity because it cannot differentiate between concurrent pathways (Fig. 5G). Meanwhile, Compass- failed to predict both MAR04042 and MAR04412 because it emphasizes the general involvement of hub metabolites (Fig. 5K), specifically aspartate and AMP in this context (Fig. 6F), which skewed the integration. By contextualizing these reactions within a pathway-integration framework, eFPA once again outperforms other methods, providing accurate predictions at both the cell and cell-type levels that the other methods only partially capture (Fig. 6G,I, Table 1).

Multiple other examples (Table 1) demonstrate that many metabolic processes in expected cell types were overlooked or failed to be properly differentiated by either reaction expression (Fig. EV4A–E), or Compass- (Fig. EV4F–K), but not by eFPA. Additionally, numerous reactions lack associated genes (Fig. 5I), with their flux potentials thus only predictable by eFPA and Compass- (Fig. EV4L–O). In these instances, eFPA was typically better at discriminating the correct cell types from others, as

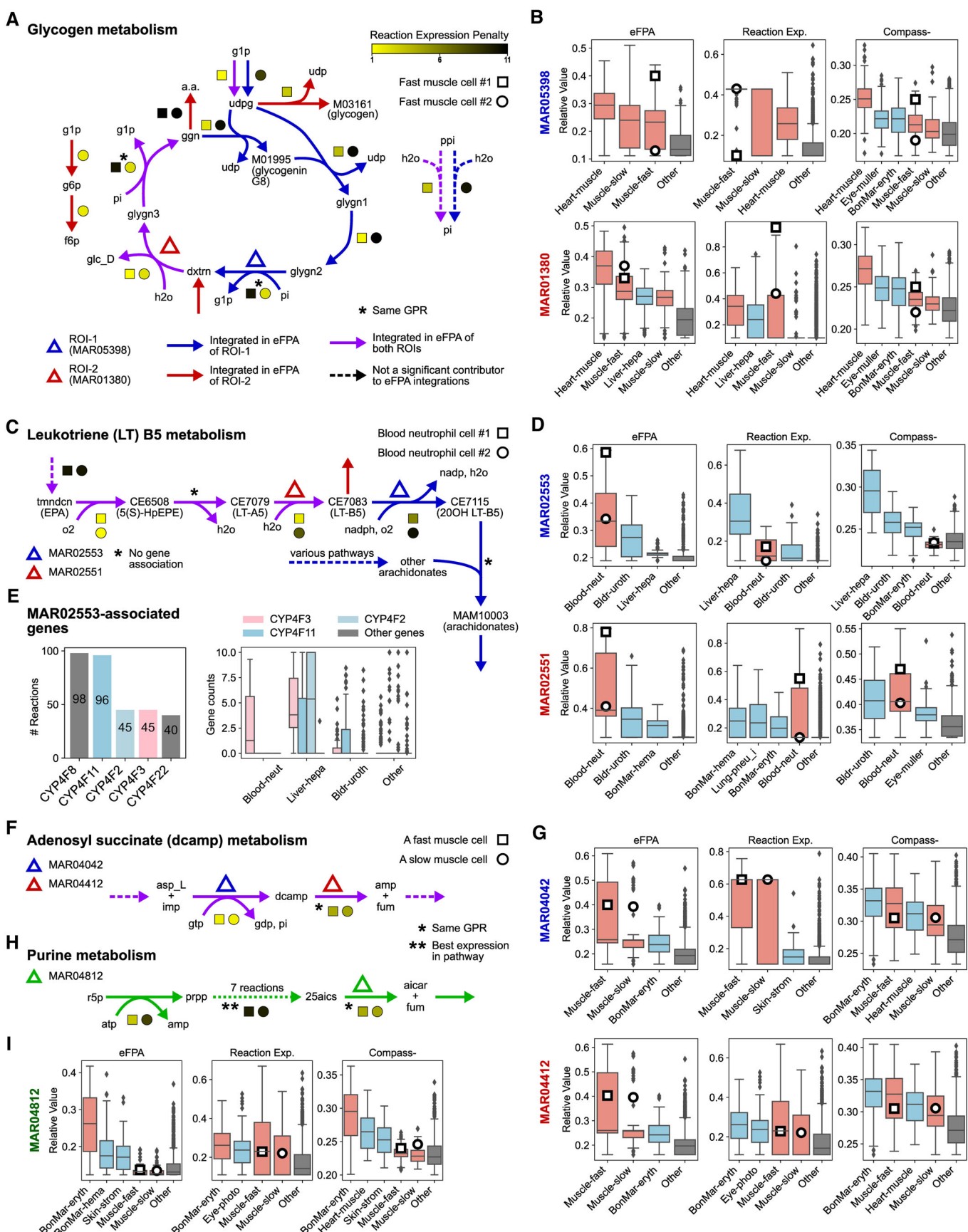

**Figure 6.  Mechanisms of prediction in single-cell data analysis.**

(A, B) Analysis of two ROIs from glycogen metabolism. (A) ROIs are depicted in the pathway diagram. Reaction expression penalty represents the weight used in flux summation for FPA and Compass- integrations. Thus, a higher value indicates lower gene expression associated with the reaction. The reciprocal of this variable corresponds to the reaction expression value used here (please see the details in Methods). Specific values for two individual cells are shown in the pathway diagram. The colors of solid reaction arrows denote the reactions whose gene expression levels were effectively integrated in the pathway selected by eFPA for the corresponding ROI. Dashed arrows represent reactions significant to Compass- integration only, using the same color scheme to indicate their relationship to the ROIs. (B) Box plots illustrate predictions of relative activity predicted from all three methods used, with cell types of interest highlighted in salmon. In addition to these highlighted cell types, the top three cell types based on activity are also shown in each plot, provided they differ from the highlighted cell types. Values for individual cells shown in (A) are represented by corresponding shapes, square and circle. (C–E) Analysis of two ROIs in Leukotriene B5 metabolism. Pathway diagram (C), box plots (D), and calculations for individual cells mirror the format described in (A) and (B). Additional charts (E) display other reactions associated with the genes of the ROI MAR02553 (bar chart) and the distribution of gene counts for each of these genes (box plot). (F–I) Analysis of two ROIs in adenosyl succinate metabolism (F, G) and one in purine metabolism (H, I). The format follows the legend of previous panels with the exception that the third ROI and related reactions in eFPA integration are indicated in green (H). The box plots in (B), (D), (G), (E), and (I) display the median (central line), IQR (box boundaries), and whiskers extending to the nearest data points within 1.5*IQR from the first and third quartiles. Points outside this range, when present, are depicted as outliers. Distributions for each cell type are illustrated with the number of data points (*n*) per cell type as specified in the table in Fig. 5A. The "Other" category includes all remaining data points not covered by the listed cell types. See Fig. 5D for a detailed example. Source data are available online for this figure.

highlighted by the example of eumelanin production in melanocytes (Fig. EV4l). Finally, there were multiple metabolic processes assigned to the correct cell type by all three methods, including glycolysis and creatine conversions in muscle cells, cardiolipin-coA formation in heart, and kynurenine metabolism in liver (data not shown). Collectively, our scRNA-seq analysis shows the robustness and effectiveness of eFPA, confirming its ability to accurately identify and predict metabolic activities in single cells.

Finally, to validate the general applicability of the improvements that were made to our integration algorithm based on yeast data, we conducted a repeat analysis using the original FPA on single-cell datasets (Fig. EV5, Table EV2). Although the PCA outcomes were similar (Figs. EV5A and 5B), likely reflecting the focus of data integration around the ROIs, FPA covered fewer cell-type-specific predictions than eFPA (Figs. EV5B and 5E). Furthermore, FPA failed to predict many metabolic activities successfully identified by eFPA, with no instances where FPA significantly outperformed eFPA (Fig. EV5C,D). For example, the activity of LT-B5 metabolism reactions MAR02553 and MAR02551 in blood neutrophils was overlooked by FPA (Figs. 5C,D and 6C,D), and predictions of dextrin synthesis (MAR05398) and degradation (MAR01380) in fast muscle cells were not as precise (Fig. 5C,D and 6A,B). These findings underscore eFPA as the most effective tool for integrating single-cell data to predict metabolic activities.

## Discussion

By systematically analyzing flux and network-integrated enzyme expression, we uncovered prevalent associations between changes in pathway-level enzyme expression and metabolic flux. This association represents an optimal balance between the analysis of enzyme levels for a ROI and a global integration that considers all enzyme levels in the network. Based on the related pathway concept, we developed eFPA, which predicts flux from enzyme expression more effectively than existing methods. The eFPA algorithm is a versatile tool for the automated interpretation of gene expression, accommodating a broad spectrum of input data, including transcriptomics and proteomics from both bulk and single-cell samples.

Our study shows that when enzyme changes in related pathways are considered collectively, broader significant associations emerge compared to focusing solely on the cognate enzyme levels of the ROI. This pathway-level association between expression and flux corroborates theoretical models of Metabolic Control Analysis (MCA), which suggests that the physiological control of flux is more likely achieved by coordinated changes in multiple enzyme levels and that the controlling effect of each individual enzyme is relatively small (Fell, 2005; Kacser and Burns, 1973). Together, the pathway-level association offers a potential reconciliation for the often-observed changes in enzyme expression with the lack of direct flux control when studying individual reactions.

While we cannot definitively prove that observed coordinated expression among enzymes in the same pathway controls flux, the practical significance lies in the demonstrated association between coexpression and relative flux levels. As a heuristic approach, eFPA uses empirically derived parameters to capture this association. However, its underlying assumptions are consistent with metabolic network architecture. Specifically, eFPA leverages the idea that enzyme expression at the pathway-level is more likely to influence the flux of a ROI than those in distant network regions. The integration of expression data across defined pathway boundaries reflects this mechanistic understanding. Although specific rules governing the related pathways of each reaction remain elusive, the use of data-driven optimization allows eFPA to capture these relationships more effectively. A key parameter learned from this optimization is the effective pathway length for integrating expression data (i.e., distance boundary). The weighted definition of distance in eFPA further optimized the integration around pathway branching points using metabolite connectivity, improving the accuracy of flux estimates.

Arguably, calibrating the eFPA algorithm using a yeast proteomic dataset may not be ideal for integrating transcriptomic data typically available for higher organisms due to differences in the type of data (proteomic vs. transcriptomic) and organism (yeast vs. animals). Nevertheless, eFPA's adaptability is supported by three factors in addition to the validations of our predictions. First, metabolic regulation in tissues is a long-term process with a significant transcriptional component (Desvergne et al, 2006), suggesting that the correlation between gene expression and flux in animal tissues may be stronger than in yeast. This is supported not only by our results with human data but also by numerous other studies where mRNA levels align with the metabolic functions of enzymes at the tissue level (Li et al, 2022; Shlomi et al, 2008). Second, our analysis of bulk human datasets demonstrates that predictions from integrating transcriptomic and proteomic data are

generally in agreement. Third, retrospective analysis of eFPA results (e.g., Table 1) indicates that, in most instances, fewer than six reactions influence the target flux calculations, which suggests that our calibrated distance boundary is appropriately set for these analyses.

Our examples with single-cell data analyses demonstrate that eFPA effectively covers predictions from both reaction- and network-level integrations, while also providing predictions missed by other methods. It is noteworthy that eFPA's pathway integration approach efficiently extracts biological insights from complex data in a way that also aligns with how researchers typically analyze pathway diagrams with overlaid gene expression (Fig. 5). A significant advantage of eFPA, particularly relevant to the growing fields of proteomics and single-cell transcriptomics, is its ability to bridge gaps in pathways. These gaps often result from sensitivity limitations inherent in both types of data, and eFPA addresses these by leveraging pathway-level integration. Together, the local associations defined in this study established an empirical foundation for interpreting metabolic gene changes observed in omics studies and eFPA provides a powerful versatile tool to efficiently translate metabolic gene changes in omics data to the predictions of flux alteration.

# Methods

**Reagents and tools table**

| Reagent/Resource | Reference or Source | Identifier or Catalog Number |
|---|---|---|
| **Software** | | |
| MATLAB 2022b | https://www.mathworks.com | |
| Gurobi Optimizer 10.0.0 | https://www.gurobi.com/ | |
| Python 3.6 | https://www.python.org | |
| cobrapy 0.26.0 | (Ebrahim et al, 2013) | |
| h5py 3.1.0/3.9.0 | https://www.h5py.org | |
| numpy 1.19.5 | (Harris et al, 2020) | |
| pandas 1.1.5 | https://pandas.pydata.org | |
| scipy 1.5.4 | (Virtanen et al, 2020) | |
| seaborn 0.11.2 | (Waskom, 2021) | |
| matplotlib 3.3.4 | (Hunter, 2007) | |
| **Other** | | |
| yeastGEM_v8.3.5 (*Saccharomyces cerevisiae* genome-scale metabolic network model [GEM]) | (Lu et al, 2019) | |
| Human1_v1.5.0 (*Homo sapiens* GEM) | (Robinson et al, 2020) | |

Brief descriptions of the main methods used in this study are included here. Details for each of the following sections are

provided in Appendix, following the same order of related sections listed here as well as in the main text.

## Processing of SIMMER dataset

Fluxomic and proteomic data from the reference study (Hackett et al, 2016) were transformed to facilitate correlation analysis and metabolic network modeling. This data processing was performed as follows:

1. We adjusted raw (absolute) flux values by dividing them by the corresponding growth rate to obtain the flux levels used throughout this study.
2. To be consistent with the flux potential analysis (FPA) (Yilmaz et al, 2020), where unscaled expression levels are used, we exponentially transformed the log2-scale proteomics data to obtain the unscaled relative abundances (referred to as *protein level*).
3. To correlate the flux of a reaction with expression level of enzymes associated with it, the *protein levels* need to be converted to a single value that represents the expression level of the reaction according to the Gene-Protein-Reaction (GPR) associations (Yilmaz et al, 2020). We followed a method described previously (Yilmaz et al, 2020) that produces a normalized, reaction-level, relative expression across conditions that varies from 0 to 1. We refer to this value as *relative expression of reaction*. The conversion function used in this procedure is available at https://github.com/WalhoutLab/eFPA/blob/master/1_yeast_modeling/FPA/scripts/calculatePenalty_partialExcluded.m. More details are stated in the Appendix Text S1.

Subsequently, the flux and protein data were mapped to the most recent consensus metabolic model of yeast (yeastGEM_v8.3.5 (Lu et al, 2019)):

1. Out of the 233 flux values, 232 were mapped to the corresponding reactions, with one reaction, r_1099, discarded because of changes in the reaction formula in the new yeast model.
2. Proteins encoded by 486 genes in the model were found to be quantified by proteomics, accounting for 42% of all model genes and associated with 809 reactions (20%).
3. Out of these 809 reactions, 658 had complete expression measurement (i.e., all associated proteins were quantified).
4. A total of 156 reactions had both the flux and enzyme expression levels determined. These reactions were used in the correlation analysis.
5. Such dual-omics dataset with fluxes and enzyme expression levels was generated for each one of the 25 conditions where media composition and dilution rate were varied.

## Correlation analysis of flux and enzyme expression level

For each of the 156 reactions whose flux and expression data are both available, we calculated the Pearson correlation coefficient (PCC) between *relative expression of reaction* and *flux* using the 25 conditions as 25 data points. The p-value of each correlation was calculated based on a two-tailed hypothesis test using the *corr* function in MATLAB 2019a and adjusted for multiple testing by *mafdr* function of MATLAB with the '*BHFDR*' method (Benjamini

and Hochberg (BH) FDR correction). The resulting FDR values were used to evaluate the correlation. A correlation was considered significant if the FDR is less than or equal to 0.05.

## Analysis of pathway-level coexpression

We derived the PCCs of *relative expression of reaction* (over the 25 conditions) for every pairwise combination of reactions in a defined pathway (Fig. EV1E and Dataset EV1) and took the median value of these PCCs to define the strength of pathway coexpression. Similarly, we defined the strength of pathway-level flux-expression correlation as the median of the flux-expression PCCs for all reactions in a pathway. Pathway-level coexpression patterns were defined as the median of *relative expression of reaction* over pathway reactions which formed a 25-element vector for each studied pathway.

## Analysis of pairwise cross-informing rate

We considered all connected reaction pairs with given flux and/or protein levels for the cross-informing rate analysis. For instance, to calculate the flux-flux cross-informing rate, we collected every connected reaction pair with fluxes determined for both reactions. We calculated PCC and p-values for each collected pair and considered reaction pairs with FDR less than 0.05 and PCC greater than 0 as *cross-informed*. The cross-informing rate was defined as the proportion of cross-informed pairs in a given set of pairs. The sets of pairs were determined based on the network degrees of bridging metabolites, i.e., metabolites that connect the paired reactions. Importantly, due to the limited number of data points (i.e., only 232 reactions with estimated fluxes), we grouped the pairs with approximate bridging metabolite degrees in the calculation of cross-informing rate. Each group produced a single data point in Fig. 2B in which the x-axis refers to the average bridging metabolite degree of pairs in the group and y-axis refers to the proportion of *cross-informed* reaction pairs, i.e., the cross-informing rate for the group.

## Enhanced flux potential analysis (eFPA)

FPA (and eFPA) is a specific flux balance analysis (FBA) problem that calculates the maximum flux potential (FP) of a ROI under certain constraints that address relative expression of reactions and their distance from the ROI (Yilmaz et al, 2020). The mathematical details of FPA can be found in Appendix Text S1 and in our previous publication (Yilmaz et al, 2020). A brief summary of this algorithm is as follows:

1. As in a regular FBA problem, FPA is done under the steady-state assumption with reaction fluxes constrained between prescribed upper and lower boundaries, which sets up the starting model.
2. Next, a weighted sum of flux in the network is constrained to be less than or equal to a constant that is referred to as *flux allowance* while the weight of each flux is inversely proportional (but not linearly, see the next section) to the *relative expression of (the corresponding) reaction*. Details on how the weight coefficients for each reaction is determined are described in the next section. This constraint sets up the

integration of expression information.

3. The flux of the ROI is selected as the objective function, which is maximized to find FP as the objective value. Since the flux allowance is a constant, reactions with smaller weight coefficients (i.e., higher relative expression) in the weighted sum are more likely to carry larger flux to maximize the flux left for the ROI, thereby conveying the influence of their expression changes to the flux potential of the ROI.

The enhanced flux potential analysis (eFPA) algorithm stays the same as original FPA except for the new distance decay functions and the use of weighted metabolic distance instead of naïve metabolic distance (see below).

## Weight coefficients and distance decay function

The key component of FPA is the calculation of the *weight coefficients*. Weight coefficient is proportional to the reciprocal of the *relative expression of reaction* and is scaled by a distance decay function. The distance decay function represents the decrease of the influence of network reactions on FP as their distance from the ROI increases. Thus, the distance function downscales the weight coefficients as the distance to the ROI increases, which would allow a reaction to take large flux values without significantly affecting target FP calculations even if its weight coefficient is high (i.e., if it is poorly expressed). The eFPA algorithm employs two redesigned distance decay functions, distinct from that in the original FPA. Please refer to Appendix Text S1 for mathematical details.

To integrate the bridging metabolite degree (an important indicator of the network architecture) (Fig. 2A) with eFPA, we weighted the metabolic distance between two adjacent reactions based on the number of connections to the metabolites that connect them (Fig. EV2B), such that the distance between reactions connected by hub metabolites is upscaled. Thus, the distance between a pair of reactions can be greater than the original distance that is the number of reactions between them plus one. We refer to the new distance measure as *weighted metabolic distance*. Weighted metabolic distance effectively encourages the influence during integration from enzymes in linear pathways to the ROI (i.e., connected by metabolites of lower network degrees), thus achieving the integration of enzyme changes of interest. The distance boundary parameter in eFPA is measured in the scale of weighted metabolic distance. To relate this parameter to the actual length of the integrated pathway for interpretation, we further converted it to an interpretable metabolic distance from the ROI (i.e., the maximum distance of integrated reactions to the ROI) that was used for data visualization in Fig. 2D and Fig. EV3. Please refer to Appendix Text S1 (sections *Mathematical formulation of Flux Potential Analysis (FPA)*, *Weight coefficients and distance decay function*, *Weighted metabolic distance* and *Calculation of effective distance boundary*) for the mathematical details.

## eFPA of SIMMER dataset

As in our previous FPA analysis (Yilmaz et al, 2020), eFPA does not rely on any quantitative exchange flux constraints as we are focused on the effect of enzyme expression changes. The model for eFPA of SIMMER dataset was constrained as follows:

1. All available nutrients based on the definition of culturing media were made freely exchangeable by setting the lower boundary of pertaining exchange reactions to -1000. These nutrients include phosphate, glucose, ammonium, uracil, and leucine (Table EV3).
2. No non-growth-associated maintenance (NGAM) was imposed (i.e., the lower boundary of NGAM reaction is set at 0) during eFPA.
3. To qualitatively account for media differences across conditions, we blocked the uptake of unavailable nutrients in each condition. For example, uracil and leucine uptakes were blocked in the eFPA of phosphate-limiting, carbon-limiting, and nitrogen-limiting conditions.
4. In addition, to address the extreme low abundance of nutrients (e.g., glucose concentration is over 20-fold lower in carbon-limiting conditions), we set an arbitrarily large weight coefficient on the pertaining exchange reactions when applicable. Please refer to Appendix Text S1 for a full list of such constraints.

For the yeast analysis, eFPA was performed by a modified version of the generic eFPA function (https://github.com/WalhoutLab/eFPA) to enable highly parallel computation in a computer cluster. The distance boundary parameter was changed according to the question of interest as indicated in the text. All 232 reactions (Dataset EV1) with determined fluxes in SIMMER dataset were analyzed with eFPA. Other details regarding parameterization, analysis and interpretation of eFPA in yeast can be found in Appendix Text S1 (sections *Correlation analysis of eFPA results and flux data*, *Titration of the distance boundary*, and *Calculation of effective distance boundary*).

## Randomization test of eFPA

To assess the statistical significance of eFPA modeling, we shuffled the rows (reaction labels) of the *relative expression of reaction* matrix (658 reactions by 25 conditions), i.e., randomized the association between the expression levels and reaction labels. After shuffling, eFPA was performed with randomized data following the same procedure as described above. This randomization was performed for 1000 times. It is noteworthy that we were unable to perform a greater scale of randomization due to the overwhelming computational demand.

## Method benchmarking using SIMMER dataset

We benchmarked the performance of eFPA against three recent algorithms, REMI (Pandey et al, 2019), Compass (Wagner et al, 2021) and ΔFBA (Ravi and Gunawan, 2021), which predict flux changes from differential gene expression data. To ensure a fair evaluation, we closely aligned the simulation setup with eFPA and optimized critical parameters for each algorithm. The specifics of the three simulations are summarized here and detailed in Appendix Text S1.

REMI was performed using the original implementation from the authors (https://github.com/EPFL-LCSB/remi), with minor adjustments for compatibility with Gurobi 10. We applied minimal model constraints, including free exchange of phosphate, glucose, ammonium, uracil and leucine. The Fold Change (FC) of protein abundance was used as the expression metric. Predicted relative fluxes were further correlated with the measured levels for evaluation. Reactions with significant positive correlation (FDR < 0.05, r > 0) were deemed as predicted. Notably, REMI involves a FC thresholding step, however, the SIMMER dataset did not include significance test to guide such thresholding. We therefore empirically evaluated various thresholds, from 1.0 (integrate all changes) to 2-fold, in 0.1 increments. We found that the number of predicted reactions was sensitive to the threshed, where 1.2 yielded the highest prediction accuracy (47 reactions predicted). To conservatively benchmark (i.e., overestimating the performance of competing algorithms), we selected 1.2 as the threshold for Fig. 3A.

Compass was originally implemented in a python package and was not compatible with user-supplied metabolic networks (Wagner et al, 2021), which prevented us from using it directly. However, the mathematical formulation of Compass closely aligns with FPA, allowing us to develop an in-house implementation of the core algorithm by modifying FPA. Specifically, we implemented the Compass score calculation following the Algorithm 2 and 3 of Compass method (Wagner et al, 2021). For consistency, we adopted the penalties used in FPA in our Compass implementation. Other configurations, such as model constraints, also remained consistent with FPA. For simulating default Compass, distance decay was disabled by using a distance order of 0. For Compass with distance decay, we used a distance order of 2.5, which yielded the best predictions in FPA (Fig. 3B), for local integration. The metabolic distance in this analysis was the unweighted, regular, metabolic distance.

ΔFBA analysis was performed using the original implementation by the authors (https://github.com/CABSEL/DeltaFBA). We followed the simulation protocol for myocyte case study in the original publication (Ravi and Gunawan, 2021), which did not use experimental fluxes as model constraints and focused only on gene expression data integration. We found that the predictive power was unexpectedly low, with only one reaction predicted. For a conservative benchmark analysis, we explored parameter optimizations of ΔFBA, including the percentile cutoff (0.05, 0.1, 0.25), model constraints (biomass production and exchange reactions), and algorithm parameters (maxflux_val: 10, 200; epsilon: 0.1, 10). However, these modifications failed to significantly enhance the prediction accuracy. We observed that some simulations terminated with non-optimal solutions (MIP gap >0.1%) after reaching the time limit (2 h). This possibly contributed to its low performance. However, it may reflect a limitation inherent to the algorithm as extremely difficult MILP that takes more than 2 h to solve is rare for the size of the yeast network model and generally impractical to use.

The source codes for the analyses with the three algorithms are available at https://github.com/WalhoutLab/eFPA.

## Processing of human tissue dataset

The quantitative proteomics and transcriptomics of human tissues contain RNA and protein levels of more than 12,000 genes across 32 normal human tissues quantified based on 201 individual primary samples (Jiang et al, 2020). We analyzed all 32 tissues using the *tissue median* provided in the referred study. We rescaled both RNA and protein tissue medians to make them suitable for system-level modeling, including the conversion of log2-scale data to unscaled values and a variance-stabilizing transformation based on Tissue Specificity score reported in the referred study (Jiang et al,

2020). Please refer to Appendix Text S3, section *Processing of tissue expression data*, for details about the data transformation.

## Human metabolic network model

The consensus human metabolic model, Human 1 (version 1.5.0), was downloaded from metabolicatlas.org (Robinson et al, 2020). To increase the numerical stability, the stoichiometry matrix of the model was adjusted to avoid reactions with overly large coefficients or small flux capacities. To model the differential nutrient availability in blood stream (Table EV4), a set of specialized uptake reactions were added to control the mass influx of each type of imported nutrient using a flux balance method that we developed previously (Yilmaz et al, 2020). Detailed descriptions on model modifications can be found at Appendix Text S3, section *Human metabolic network model*.

## eFPA analysis for human tissues

To comprehensively model the tissue metabolism in humans we followed the modeling pipeline MERGE, which we previously developed for modeling *C. elegans* tissue metabolism (Yilmaz et al, 2020). The procedure is summarized as follows:

1. A semi-quantitative modeling of on/off status and direction of reaction fluxes was performed using iMAT++ algorithm (Yilmaz et al, 2020). This step is to globally fit the distinct expression levels of enzymes in tissues to the metabolic network, to derive:
    a. A flux distribution that we named as Optimized Flux Distribution (OFD) and that can be used to assign the flux directions for reversible reactions.
    b. A tissue-specific metabolic network in which inactive reactions that cannot carry flux according to flux variability analysis (Yilmaz et al, 2020) were removed from the network.
2. The eFPA analysis was performed on the derived tissue networks.
3. The rFP predictions generated from eFPA was overlayed with OFD to derive high-confidence predictions with reaction directionality. This step is important for the interpretation of eFPA predictions of reversible reactions that show high flux potential in both directions, such that the direction predicted by iMAT++ is chosen as the most likely metabolic function with high potential.

Together, this tissue modeling pipeline provides a comprehensive collection of high-quality predictions about tissue-enriched metabolic fluxes. In this analysis, two sets of tissue networks were built for the 32 tissues based on either proteomics data (protein-based tissue network; PTN) or RNA-seq data (RNA-based tissue network; RTN). PTN and RTN were used in eFPA based on proteomics and RNA-seq data, respectively. The decay function of eFPA algorithm used in human tissue modeling was the exponential decay with a distance boundary of 6 (i.e., the standard eFPA was used), and the distance measurement is the weighted metabolic distance. When not specified (i.e., referred to as "eFPA" or "Protein eFPA"), the input data for eFPA were protein levels of genes that are commonly detected by both proteomics and RNA-seq (12,121 genes, covering 2869/3627, 79%, genes in the Human 1 model). For eFPA using RNA expression as input (referred to as

"RNA eFPA" in figures and texts), we performed eFPA using either commonly detected genes (12,121, labeled as "common genes" or not specified) or all detected genes by RNA-seq (19,273, labeled as "all genes").

To derive the tissue-metabolic landscape, we analyzed all regular reactions in human 1 model except for transport, exchange, and custom uptake reactions (see above). To model the tissue-enriched metabolites, we performed eFPA on transporter reactions that take up or secrete metabolites from or into extracellular space, respectively. The computation was performed in Massachusetts Green High Performance Computing Center (GHPCC) with 512 cores. To exemplify the computational performance, eFPA of all internal regular reactions (7248) and 32 tissues took less than 30 min.

Detailed parameters, settings and modifications can be found in Appendix Text S3, sections from *Building human tissue networks* to *eFPA analysis: regular reactions*.

## Clustering rFPs of internal reactions

To generate Fig. 4B, reactions were filtered unless their ΔrFP were greater than 0.2 (tissue-enriched) or lower than −0.2 (tissue-depleted) in at least one of the 32 tissues. To address the cases when eFPA predicts tissue enrichment for both directions of a reversible reaction, we selected the eFPA predictions based on the relevant flux distribution (i.e., OFD) predicted by iMAT++ (see above). The reactions that passed these criteria were merged at the end. We termed these final filtered predictions *high-confidence predictions of human tissue metabolism* and generated the clustergram with them using the *clustergram* function in MATLAB with 'cosine' distance for both row and column clustering. These high-confident predictions were starred (i.e., labeled with "*") in their reaction ID in Dataset EV4.

## Clustering tissue-enrichment of subsystems

To visualize the subsystem-level metabolic specialization in tissues, we generated a matrix where rows are subsystems and columns are the 32 tissues. The values in the matrix are number of tissue-enriched reactions (ΔrFP greater than 0.2) assigned to each subsystem and tissue. The matrix was row-wise normalized by dividing all values with the maximum, which yielded the relative tissue-enrichment for subsystems shown in the heatmap (Appendix Fig. S3).

## Analysis of single-cell data with eFPA

The single-cell dataset was extracted from the Tabula Sapiens data matrix (Tabula Sapiens et al, 2022), which was obtained in hierarchical data format (HDF). We selected a total of 23 cell types from the dataset. This selection was strategically aimed at diversifying the range of organs available and included cell types from the four main tissue categories (epithelial, connective, nervous, and muscle). To ensure a manageable and balanced dataset, we randomly selected 100 cells from each cell type. An exception was made for melanocytes, where only 64 cells were available, and this cell type was specifically chosen due to its known distinct function (Lin and Fisher, 2007). It is important to note that neurons are underrepresented in the original dataset, with only eye

photoreceptors and Muller cells representing this system in our selection.

We integrated this data with the Human 1 metabolic model, the same model used in the previous analysis, but with all reactions that carried no flux removed to reduce the computational burden. These non-functional reactions were identified using flux variability analysis (Mahadevan and Schilling, 2003). Our study focused exclusively on predicting the flux potentials of cytosolic reactions, which served as ROIs, while the entire metabolic network was used with default constraints during the integration process.

The single-cell integration algorithms included standard eFPA, reaction expression, and Compass- (see Appendix Text S4 for details). Reaction expression calculates the *relative expression of reaction* (see above) for each ROI in every cell, corresponding to the reciprocal of the ROI expression penalty. This penalty serves as the weight coefficient of the ROI used in eFPA. Standard eFPA predicts flux potentials as detailed previously. Compass- calculates flux resistance as described in the original publication (Wagner et al, 2021), and it can use either the penalties as specified in eFPA or those outlined in the original Compass study. Our evaluations of Compass- with both penalty systems showed a slight improvement when using eFPA penalties (data not shown); hence, this configuration was adopted to ensure a fair comparison across different algorithms. Additionally, Compass- resistance scores were normalized to range between 0 and 1, aligning with the scaling method used in the other algorithms. For eFPA and reaction expression, the flux potential or reaction expression of the ROI in a single cell is divided by a theoretical maximum, represented by a hypothetical 'super system' where all genes are expressed at their maximum observed level (Yilmaz et al, 2020). Similarly, for Compass-, the resistance score of a super cell is used as the denominator to scale the resistance score of the ROI, yielding a relative Compass- score that is comparable to those derived from eFPA and reaction expression. The "minus" (−) in the name of the Compass- algorithm signifies the omission of cell lumping, a common technique used to mitigate noise in single-cell data. As a result, scores from all methods reflect the relative capacity of each ROI to carry flux in a given cell, based solely on its transcriptome.

The cell-type enrichment of predictions from each algorithm was assessed using multiple methods. First, we calculated the median fold change (FC) and obtained p-values from a Wilcoxon rank-sum test between the ROI scores of a specific cell type and other cell types (e.g., Fig. 5D). To prevent bias in our comparative analysis, we excluded the highest ranking cell type from each comparison, particularly considering the dominant influence of highly metabolically active tissues like the liver. Additionally, to maintain consistency when evaluating the top-ranking tissue itself, we excluded the second highest ranking cell type. This methodology ensures a balanced evaluation across all cell types by preventing any single tissue from overshadowing others due to its metabolic prominence. Second, we evaluated the rank of each cell type with respect to the medians (Table 1). Third, we visualized the relationship of each cell type to others using box plots (e.g., Fig. EV4), arranging cell type scores in decreasing order. These plots consistently included the top three cell types and those of particular interest, selected based on established knowledge about the metabolic function of the ROI. The combination of box plots in

Figs. 6 and EV4 with the statistical data in Table 1 provided a comprehensive overview of the performance of each method across the analyzed cases. Finally, we repeated the single-cell analysis using the original FPA algorithm to directly compare its results with those from eFPA, as presented in Fig. EV5 and Table EV2.

## Blinding

No blinding was done in this study as it involves computational analysis of predefined datasets.

## Data availability

The datasets and computer code produced in this study are available in the following databases: Modeling computer scripts: GitHub (https://github.com/WalhoutLab/eFPA), Input created for single-cell data analysis: Zenodo (https://doi.org/10.5281/zenodo.13801228).

The source data of this paper are collected in the following database record: biostudies:S-SCDT-10_1038-S44320-025-00090-9.

## Peer review information

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

## Acknowledgements

We thank members of the Walhout lab for supports and discussions on this project. We thank Caryn Navarro, Hefei Zhang, Job Dekker, Olga Ponomarova, Robert Brewster, and Shivani Nanda for discussions on the manuscript. This work was supported by a grant from the National Institutes of Health GM122502 to AJMW.

## Author contributions

**Xuhang Li**: Conceptualization; Data curation; Software; Formal analysis; Validation; Investigation; Visualization; Methodology; Writing—original draft; Writing—review and editing. **Albertha J M Walhout**: Conceptualization; Supervision; Funding acquisition; Project administration; Writing—review and editing. **L Safak Yilmaz**: Conceptualization; Data curation; Software; Formal analysis; Supervision; Validation; Investigation; Visualization; Writing—review and editing.

Source data underlying figure panels in this paper may have individual authorship assigned. Where available, figure panel/source data authorship is listed in the following database record: biostudies:S-SCDT-10_1038-S44320-025-00090-9.

## Disclosure and competing interests statement

The authors declare no competing interests. Albertha JM Walhout is an editorial advisory board member. This has no bearing on the editorial consideration of this article for publication.

# Expanded View Figures

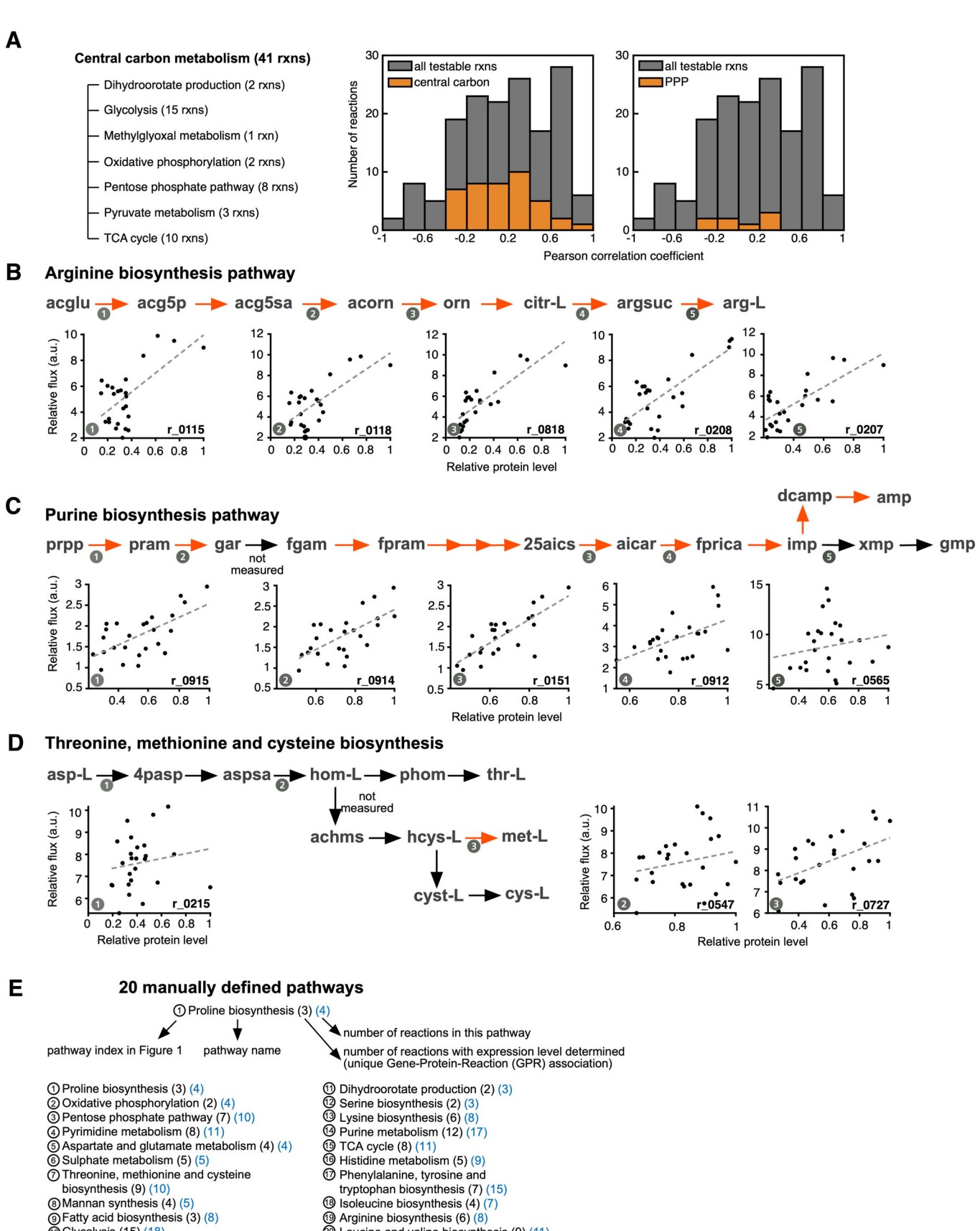

**Figure EV1.   Analysis of flux-enzyme level correlation in metabolic pathways.**

(A) The PCC distribution for reactions of central carbon metabolism. The distribution of reactions of the pentose phosphate pathway (PPP) is shown on the right. (B–D), overlay of the flux-enzyme level correlations on selected metabolic pathways. The three selected pathways include ones that are fully (B) or partially (C) composed of correlated reactions, and one that has only one correlated reaction (D). Orange arrows indicate reactions that show significant correlation (FDR < 0.05, PCC > 0) and black arrows indicate uncorrelated reactions. (E) Twenty manually-defined pathways that are labeled by their indices in Fig. 1E, F.

## A

Flux Potential Analysis

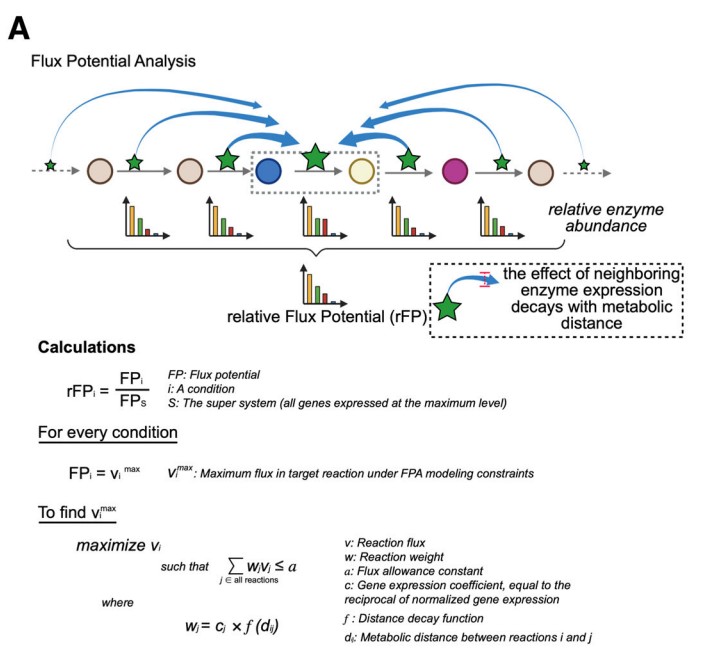

**Calculations**

$$rFP_i = \frac{FP_i}{FP_s}$$

FP: Flux potential
i: A condition
S: The super system (all genes expressed at the maximum level)

For every condition

$$FP_i = v_i^{\,max}$$

$v_i^{\,max}$: Maximum flux in target reaction under FPA modeling constraints

To find $v^{max}$

maximize $v_i$

such that $\sum_{j \in \text{all reactions}} w_j v_j \leq a$

v: Reaction flux
w: Reaction weight
a: Flux allowance constant
c: Gene expression coefficient, equal to the reciprocal of normalized gene expression

where

$$w_i = c_i \times f(d_{ij})$$

f : Distance decay function
$d_{ij}$: Metabolic distance between reactions i and j

## B

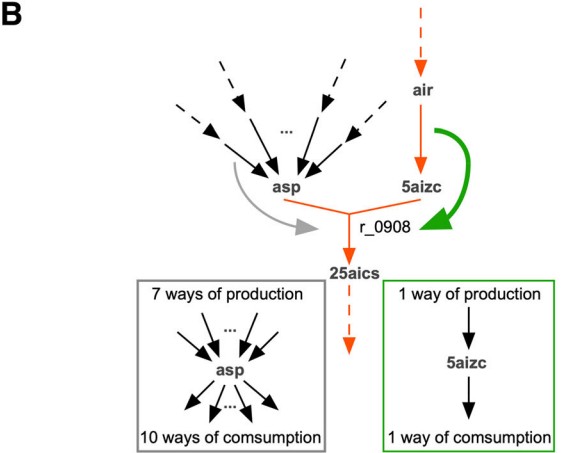

Weighted Distance = (7 + 10) / 4 = 4.25
Weighted Distance = (1 + 1) / 4 = 0.5

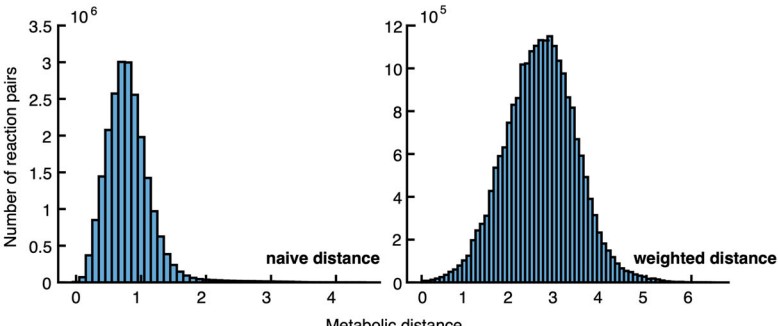

## C

binary decay function: $f(d_{ij}) = \begin{cases} 1, & d_{ij} \leq n \\ 0, & d_{ij} > n \end{cases}$

exponential decay function: $f(d_{ij}) = \frac{1}{1+2^{d-n}}$

n: distance boundary

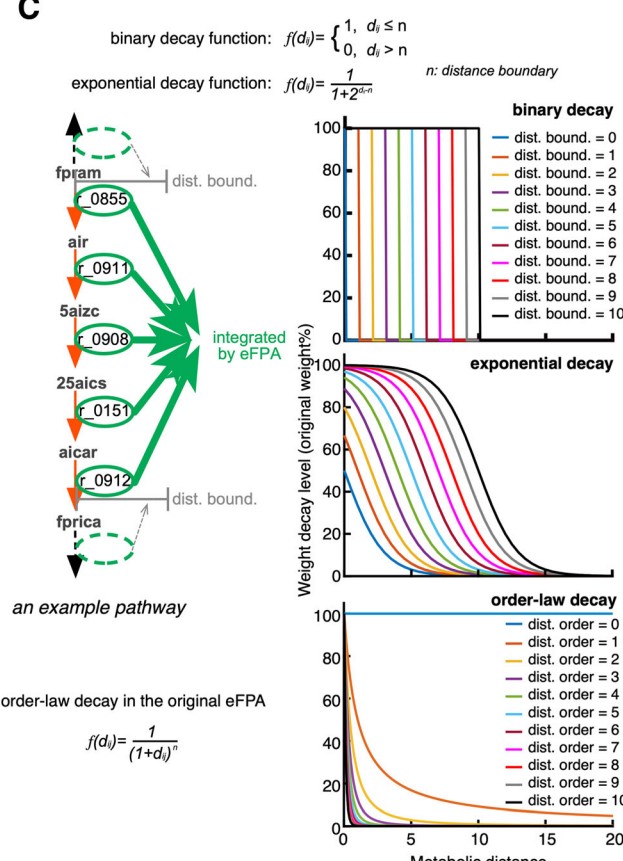

order-law decay in the original eFPA

$$f(d_{ij}) = \frac{1}{(1+d_{ij})^n}$$

**Figure EV2. Principles of FPA and eFPA.**

(A) Schematic of flux potential analysis (FPA). FPA and eFPA integrate the levels of enzymes that catalyze reactions surrounding a ROI to predict relative flux potential (rFP) of the ROI. The contribution of surrounding enzymes can be tuned by a distance decay function such that FPA is versatile to integrate expression information from a local subnetwork to the entire network. The mathematical formulation of FPA is briefly summarized (also see Appendix Text S2). (B) The weighted metabolic distance. A cartoon illustrating the calculation of weighted metabolic distance is shown on the top. On the bottom, the distributions of naïve (unweighted) and weighted metabolic distance are shown for all reaction pairs in the yeast metabolic model. (C) The distance decay functions of eFPA. The formulas of binary (used in optimal-boundary eFPA) and exponential decay functions (used in standard eFPA) are shown in the figure. The left panel shows a cartoon to illustrate the concept of distance boundary. The right panel shows the decay curve of the two functions.

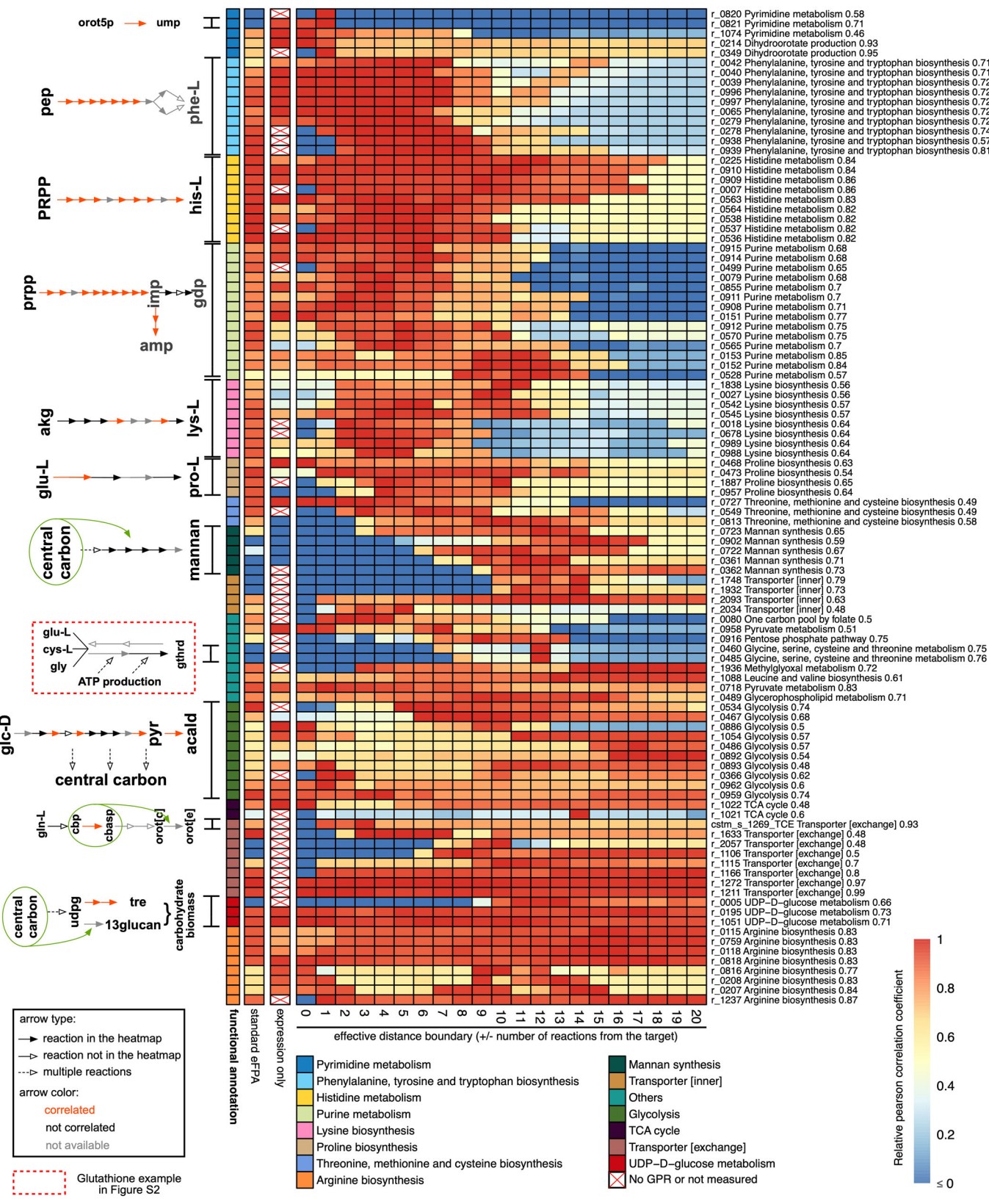

**Figure EV3. Heatmap of PCC between flux and rFP with a titration of distance boundary.**

The relative PCC (row-wise normalized by dividing each value by the row maximum) for 102 predicted reactions. The numbers on x-axis indicate the effective distance boundary, which represents the calculated actual length of the integrated pathway based on the distance boundary parameter in the scale of weighted metabolic distance. ROIs (rows) are arranged based on their position in the pathway they are associated with, and reaction IDs are indicated on y-axis together with the associated pathway and the optimal-boundary PCC. Significant contributors (e.g., relevant pathway reactions) to eFPA analysis of some ROIs are depicted on the left.

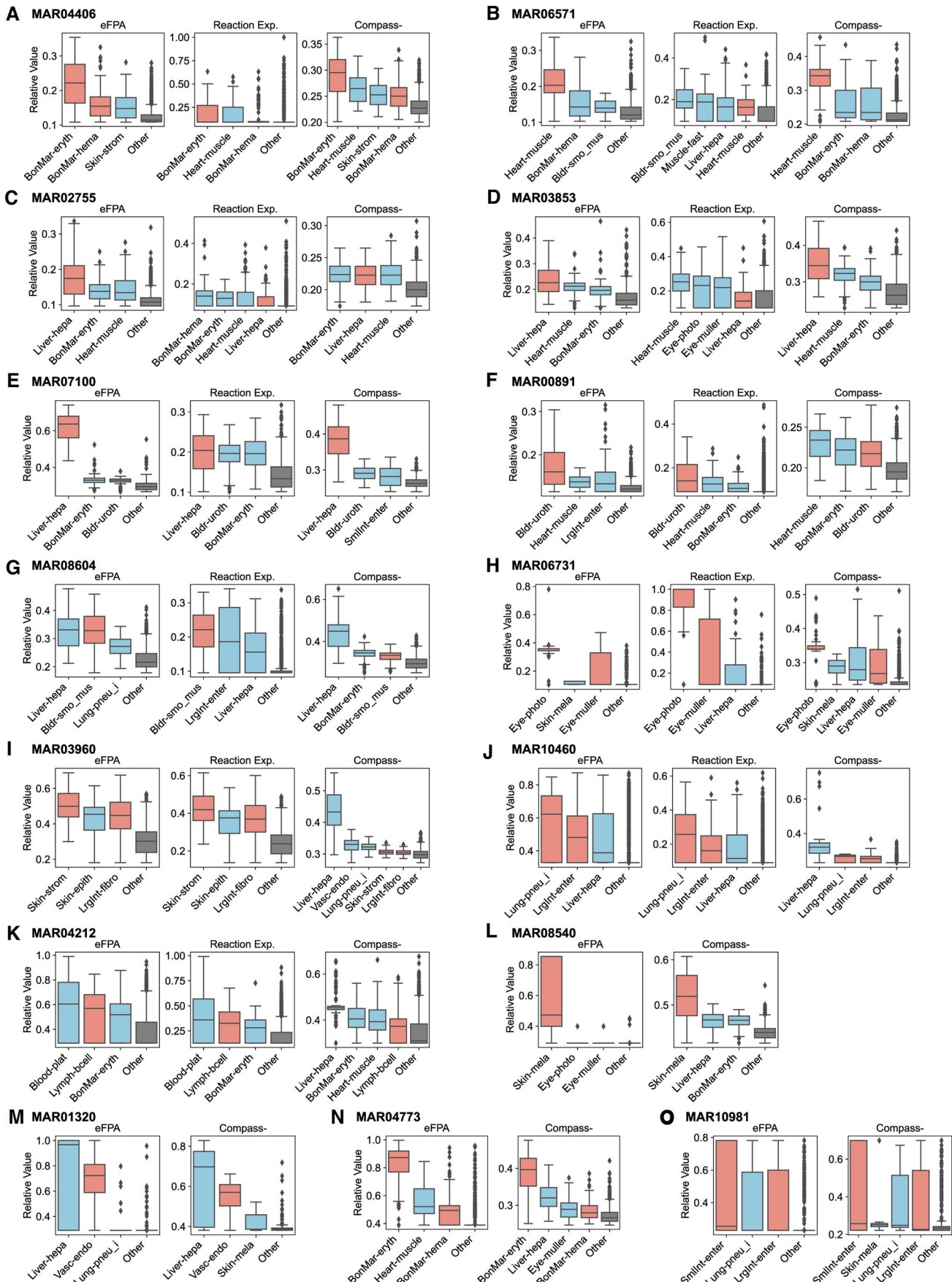

**Figure EV4.  Verifiable cell-type-enriched cytosolic reactions in Human 1.**

Examples in Table 1 are displayed as box plots, with cell types of interest highlighted in salmon. In addition to these highlighted cell types, the top three cell types based on the predicted flux activity are also shown in each plot, unless they overlap with the highlighted cell types. (**A–K**) Examples where reaction expression or Compass- fails to clearly differentiate one or more of the referred cell types of interest from Table 1. (**A**) The activity of a primary purine biosynthesis reaction is not well-differentiated by reaction expression in erythroid progenitors and is missed by both reaction expression and Compass- in hematopoietic stem cells. (**B–D**), Activities of indicated reactions are missed by reaction expression in the cell type of interest (Table 1). (**E**) Xenobiotic detoxification reaction activity in hepatocytes is poorly differentiated by reaction expression. (**F, G, I–K**) Activities of indicated reactions are missed by Compass- in the cell type(s) of interest (Table 1). (**H**) Dopamine production potential in Muller cells is comparatively low in Compass- evaluations, especially against hepatocytes and melanocytes. (**L–O**) Activities of reactions without gene associations, predicted only by eFPA and Compass-. While both methods generally provide accurate cell-type enrichments, eFPA typically offers better differentiation of the expected cell types (Table 1). All box plots display the median (central line), IQR (box boundaries), and whiskers extending to the nearest data points within 1.5*IQR from the first and third quartiles. Points outside this range, when present, are depicted as outliers. Distributions for each cell type are illustrated with the number of data points (*n*) per cell type as specified in the table in Fig. 5A. The "Other" category includes all remaining data points not covered by the listed cell types. See Fig. 5B for a detailed example.

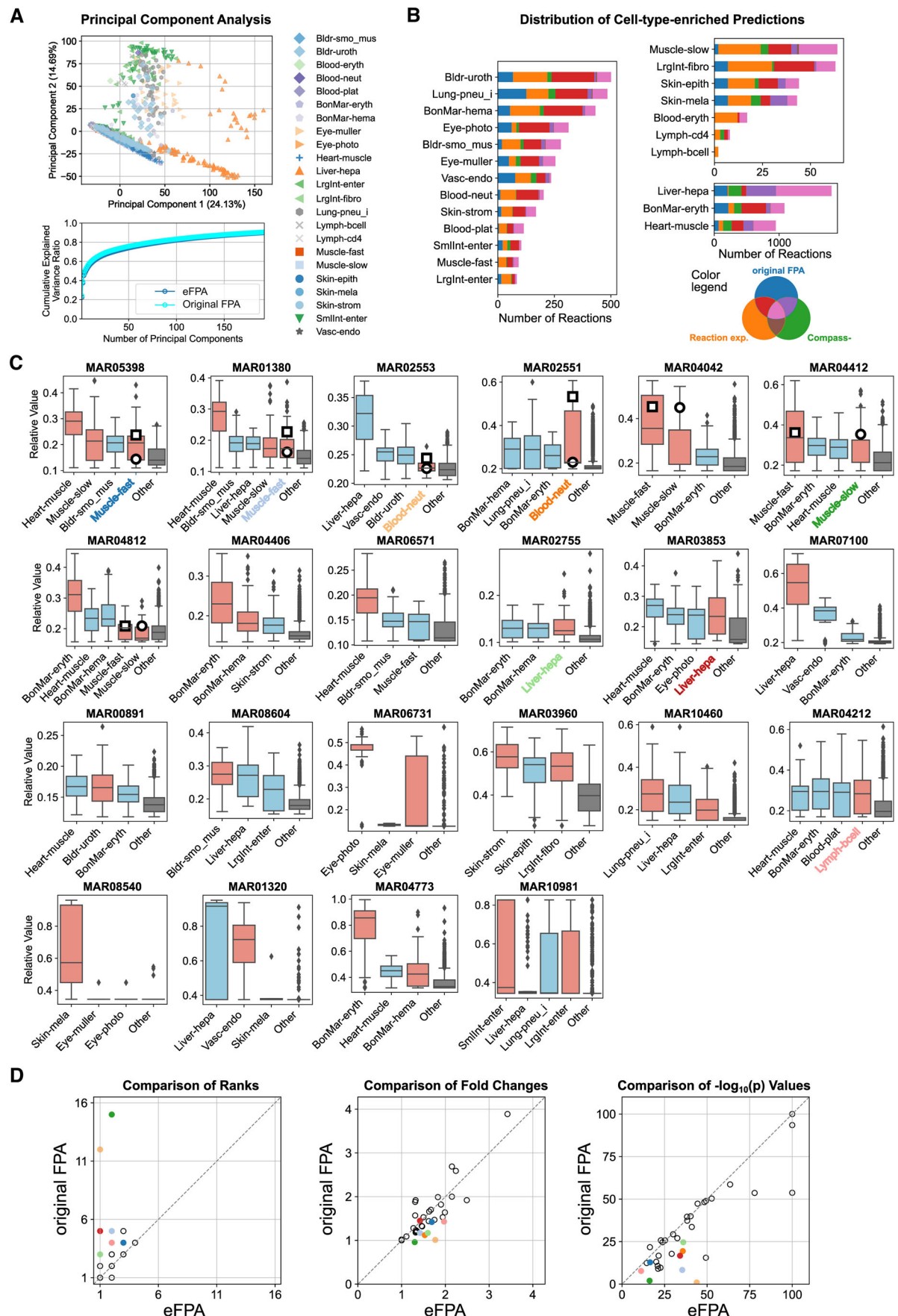

◀ **Figure EV5.  Integration of single-cell data with original FPA to enable comparison with eFPA results.**

(A) PCA of predicted relative reaction flux potentials. Top panel shows Clustering of single cells using first two principal components (as in Fig. 5B); bottom panel shows cumulative variance explained by principal components (as in Fig. 5C). Percent variance explained is indicated on the axes (top panel). (B) Distribution of cell-type-enriched reactions for each cell type (as in Fig. 5E). (C) Box plots of FPA predictions (as in Figs. 6B, D, G, I and EV4), with cell types of interest highlighted in salmon. Colored cell types in x-axis labels match colored data points highlighted in (D). The plots display the median (central line), IQR (box boundaries), and whiskers extending to the nearest data points within 1.5*IQR from the first and third quartiles. Points outside this range, when present, are depicted as outliers. Distributions for each cell type are illustrated with the number of data points (*n*) per cell type as specified in the table in Fig. 5A. The "Other" category includes all remaining data points not covered by the listed cell types. See Fig. 5D for a detailed example. (D) Comparison of predictive power metrics (rank, fold change, and *p*-values; Table 1, Table EV2) between eFPA and FPA. Colored data points indicate predictions where eFPA significantly outperforms FPA, with colors matching those in (C). In the log-transformed *p*-value plot, values above 100 are capped at 100 for visualization.

