## [Peer Review File · Molecular Systems Biology]

Enhanced flux potential analysis links changes in enzyme expression to metabolic flux

Xuhang Li, Albertha Walhout, and Lutfu Yilmaz

Corresponding author(s): Lutfu Yilmaz (LutfuSSafak.Yilmaz@umassmed.edu) , Albertha Walhout (marian.walhout@umassmed.edu)

Review Timeline:

Submission Date:	28th Aug 23
Editorial Decision:	9th Oct 23
Appeal Received:	20th Sep 24
Editorial Decision:	5th Nov 24
Revision Received:	31st Dec 24
Editorial Decision:	9th Jan 25
Revision Received:	29th Jan 25
Accepted:	7th Feb 25

Editors: Maria Polychronidou and Poonam Bheda

Transaction Report:

9th Oct 2023

RE: Manuscript MSB-2023-11970, Linking enzyme expression to metabolic flux

Dear Marian,

Thank you again for submitting your work to Molecular Systems Biology. We have now heard back from the three referees who agreed to evaluate your study. As you will see below, the reviewers raise substantial concerns on your work, which unfortunately preclude its publication in Molecular Systems Biology.

The reviewers appreciate that the study addresses a relevant topic. However, they are not convinced that the study presents a decisive advance and they raise several technical concerns related to the performed analyses. During the cross-commenting process, in which the reviewers get the chance to make additional comments based on each other's reports, the reviewers indicated that they all seem to be in alignment in that they do not find the study well suited for publication in Molecular Systems Biology.

Taken together and given the substantial concerns raised, I am afraid that we cannot offer to publish the study. I am sorry that the review of your work did not result in a more favorable outcome on this occasion, but I hope that you will not be discouraged from submitting future work to Molecular Systems Biology. In any case, thank you for the opportunity to examine this work.

Kind regards,

Maria

Maria Polychronidou, PhD
Senior Editor
Molecular Systems Biology

Reviewer #1:

This theoretical study based on correlation combines data on proteomics-derived enzyme levels and intracellular fluxes from a single yeast publication to ask how well protein expression data predict which fluxes, which so far is considered to explain only on the order of 1/3 to 1/2 of the fluxes in metabolism. Some seemingly moderate adjustments are needed for the metabolic correlation analysis. A rather broad sweeping claim is made for three "paradigms" of enzyme levels being associated with fluxes through the immediate reaction, the pathway, and some other coupled reactions that are further away. Making several simplifications and strong assumptions by replacing essentially every data type used for yeast with something else, the MS ends with a short section on trying to make a similar case for human metabolism. Overall, I miss biological insights/novelty and remain skeptical about the relevance of the 3 paradigms. Some observations correlate but the advance over previous work is not clear, causality is not established, and there is no attempt to actually test the conclusions, which in some cases are even wrong as outlined below.

Main points:

1. The 3 paradigms are not new. Nobody doubts that enzyme levels influence the catalyzed reaction, the pathway flux and in some cases more distant fluxes (ie flux coupling), the latter being a well-known and integral element of flux balance analysis. Previous work did find some but not very strong correlation of enzyme/mRNA levels with catalyzed reactions and the local pathway flux. Although no quantitative comparison is offered in the present MS, it seems that somewhat more enzyme-flux correlations are found. My concern is that the reported results are more or less the same as in all previous studies in that enzyme levels correlate with/explain some fluxes to some extent. For yeast it is well known that for example respiration and TCA cycle are mainly transcriptionally controlled, presumably because of the major protein investment.

The main difference of the present work is that the authors include, beyond central metabolism, also about 100 biosynthetic reactions for which fluxes can be easily inferred from biomass formation. These biosynthesis reactions are the simpler part that again is well known to be under strong genetic control, in particular in the used data set where amino acids were used for perturbation and where the increasing growth rate in chemostats depends in a linear fashion on higher flux through biosynthesis reactions. There is a danger that the chosen data sets and the focus on biosynthesis reactions (about 2/3 of all in the chosen network and manually defining 20 synthesis pathways for pathway correlation; line 9 p 7) introduces a strong bias. Along these lines I believe it is a misconception that "enzyme usage significantly impacts flux in amino acid biosynthesis" with reference to other work (P 6). If more biosynthesis is necessary biosynthesis genes are expressed stronger - which still does not mean that

they control or cause the flux through a given reaction. Some correlation with the biosynthesis flux is of course expected and known, but it does not explain the flux in any quantitative sense.

Overall, it remains unclear what the really novel contribution of this work is and how much better it explains what?

2. The enzyme reach aspect seems to be primarily a reinvention of the well-known flux coupling. Before it can be claimed to be a "novel" parameter, it would need to be formally checked against the general flux coupling that is inherent to metabolic networks. My expectation is that enzyme reach is only a subset of generally flux coupled reactions. The claim that this work "revealed" that enzymes can influence also distant reactions (eg start of discussion and elsewhere) is not justified by any means. This has long been known.

3. By remaining strictly at the level of correlations, the work sidesteps the more relevant causality. To make the work meaningful it would be important to explain how the newly found correlations help to overcome the general consensus in the field that fluxes are NOT controlled by expression, which is based on several key observations such as i) enzymes typically operate well below their in vitro K_m and V_{max} ; ii) enzyme overexpression typically does not increase flux (neither in the reaction nor the pathway); iii) modulating enzyme levels up/down over quite a range does not affect flux; iv) the timing of expression does not match most metabolic responses in microbes such as yeast; and v) expression changes were repeatedly shown to be poor predictors of what cells need in a given conditions (Price, Deutschbauer et al 2013 *MSG* or Birrell, Brown et al 2002 *PNAS* and many others).

Moreover, alluding to MCA in the discussion that this theoretical concept of shared flux control would be supported by the present analysis is a misconception. MCA is a great theoretical framework but works only for infinitesimally small changes in expression upon which the control would immediately shift to another reaction - not for the huge steady state values considered here.

4. There are only very few biological interpretation attempts of observed correlations and in some cases they are wrong. On page 10 the correlation of fluxes to glutathione with those through glycolysis, ox phos and TCA cycle is claimed to suggest that glutathione synthesis would be energy limited. Firstly, there is no reason whatsoever why the synthesis of this tripeptide should be energy-limited. Second, the situation is more likely be the reverse. Increased respiration generates more ROS, which in turn requires more defense molecules such as glutathione - a long and well-known relationship. Besides the obviously wrong interpretation, this illustrates the problem with having only correlations.

5. The final section on human metabolism is only possible by making sweeping assumptions such as replacing protein with mRNA data and reaction fluxes with uptake rates. These are completely novel input data that require lots of (also partly questionable) changes and assumptions that are only dealt with as an afterthought. The claim that mRNA levels can be reliably used instead of proteomics goes against a huge body of evidence suggesting that the two quantities are not well correlated. How is it then possible that it works just as well in eFPA and what does it mean for that analysis? In general, this section is not well supported beyond a superficial analysis and is probably better removed.

The test itself in this section is also not clear. What does it mean that "... we could validate boundary flux predictions with available tissue-enriched metabolite annotations" (P 13 L 12/13)? An annotation is only the name of a metabolite? It is also a major restriction to have only uptake and not intracellular fluxes, all of which is not well explained and discussed in terms of consequences.

Minor points

1. Text contains many generalisms and unclarities. Here are a few examples:

- p 2 bottom: the artificial separation of expression mechanisms in innate (during development) and in response to conditions. Even developmental changes are based on exogenous factors, besides it would no sense for the here studies yeast anyway. From this rewiring of networks is separated from wiring. This is not substantiated by any evidence.

- unclear what "flux data adjusted to growth rates" are. The usual concept is to normalize to the uptake rates so that relative fluxes are obtained.

2. On P 9 the reaction network has 232 reactions while on P 5 it is 156?

Reviewer #2:

In the manuscript, "Linking enzyme expression to metabolic flux," Dr. Walhout and colleagues examine the extent to which enzyme expression levels are correlated to flux through metabolic pathways when network architecture is accounted for. Though previous studies have shown that metabolite availability predicts reaction fluxes better than enzyme availability, the authors note that previous work has emphasized individual reaction fluxes, rather than taking a network approach. Using a previously published data set of reaction fluxes and enzyme levels in yeast under various environmental conditions, they find that if enzymes regulating closely related pathways are co-expressed, there is a stronger correlation between enzyme levels and

flux. This result suggests that enzymes have a "reach" of influence within a network, which extends beyond the specific reactions they catalyze. The authors incorporated this concept of enzyme reach into a previously published algorithm (FPA), and applied this modified algorithm (eFPA) to predict flux values from enzyme expression levels in both yeast and human tissues.

Critiques

The central concept of the paper is interesting, but the paper lacks a clear rationale...what does the result imply about the coupling or regulation of enzymatic flux? How could approaches for integrating expression data into network models be improved given the results presented?

If we assume that the results of the paper allow for better integration of expression data into network models (e.g., enzyme expression that correlates with flux could be handled differently than enzymes that don't correlate with flux), what kind of concerns are there with the fact that the analyses are based on a single data set? How generalizable are the results so that those concepts could be applied to another system for which there isn't coupled expression and flux data?

The number of new concepts (enzyme reach, eFPA, etc.) makes the paper very difficult to understand. Similarly, there are so many "co" words (correlated, collinear, coupled, co-expression, connectivity); it's hard to keep track of what they mean in the context of this specific study. Relatedly, the figures are extensive and detailed, but at the expense of interpretability. Text is very tiny (especially EEV4 and EV5) and there's just so much data and analyses in each of the 4 main figures, each of the 5 extended figures, and all the supplement. It seems the work would be better suited split up into a couple papers or streamlined even more so that the essence of the messages aren't lost.

In human tissues, how do eFPA predictions of metabolite fluxes from mRNA compare to other methods of predicting fluxes that don't include enzyme reach? Are there key differences or additional insights that this method offers?

Page 3, lines 19-23 seem to conflict with the assertion made on page 6, lines 22-23

Page 8 - briefly explain the concept of metabolite degree. For example, "Metabolite degree captures the total number of sources and sinks for the metabolite..."

Figure 2A - typo ("consumption")

Figure 3a - don't understand the difference between the black line and blue line

Reviewer #3:

In this study the authors aim to get insight into how flux is controlled in metabolic networks. The authors state that previous studies demonstrated that flux is controlled by metabolite concentration rather than by enzyme concentration, and here the authors proposed the control of enzymes only on fluxes of non-cognate reactions. The authors use data from yeast in their study and in the last part of the paper they extrapolate their findings from yeast also to human metabolism. Overall the finding in this study is interesting but there are quite a few terminologies not appropriately explained in the manuscript, preventing full understanding and assessment of the content. Please consider the comments below.

What is the biological basis of the hypothesis that enzyme levels affect metabolic flux of non-cognate reactions? In other words, if a good correlation is observed between an enzyme and flux of a non-cognate reaction, by which biological mechanism the enzyme controls the non-cognate reaction?

New terminologies should be clearly defined, especially cognate reaction and enzyme reach.

It would be better to demonstrate whether and how much eFPA outperforms the original FPA.

P8 L20-23: does FPA calculate relative flux potential (rFP) or flux potential (FP)? What is the output of FPA?

P9 L14: what is "the corresponding eFPA"? Is eFPA here the algorithm itself or an output (e.g. maybe rFP or FP) of this algorithm?

P10 L7: Fig. 2g appears to indicate that levels of metabolic enzymes do not affect their cognate reactions.

Fig. 1e: in the upper box all the five enzymes are linked together but in fact they should be just pairwise linked. Is this also the same issue in Fig. 1f?

Legend of Fig. 3a: "The number of reaction fluxes predicted by either collinearity or eFPA are shown in red". 1) Collinearity is defined as PCC between two fluxes (P11 L9) but how does it predict fluxes? 2) While it is not clear eFPA predicts rFP or FP,

both rFP and FP are different from reaction flux.

Fig. 3bc: what is the relative flux? Is it rFP?

** As a service to authors, EMBO Press offers the possibility to directly transfer declined manuscripts to another EMBO Press title or to the open access journal Life Science Alliance launched in partnership between EMBO Press, Rockefeller University Press and Cold Spring Harbor Laboratory Press. The full manuscript and if applicable, reviewers' reports, are automatically sent to the receiving journal to allow for fast handling and a prompt decision on your manuscript. For more details of this service, and to transfer your manuscript please click on Link Not Available. **

Response:

We thank the editor and reviewers for their thorough and constructive review. We have carefully considered the comments and acknowledge the concerns raised, particularly regarding our initial conclusions about flux regulations based solely on correlation analyses of a single dataset. To address these issues, we have restructured our manuscript into a method development paper. In this revised version, we refrain from making definitive statements about flux regulation. Instead, we focused on elucidating the association between changes in flux and enzyme levels.

To enhance our work as a method development study, we include benchmark analyses against competing methods and introduced a new section that demonstrates the application of our method to single-cell RNA-seq (scRNA-seq) data. While the presentation of our key findings has been modified, we believe these changes have significantly improved the scientific rigor of our study and have better highlighted its novelty.

We trust that the revised manuscript is now suitable for publication in *Molecular Systems Biology*. We have responded to all original comments point-by-point in blue.

Reviewer #1:

This theoretical study based on correlation combines data on proteomics-derived enzyme levels and intracellular fluxes from a single yeast publication to ask how well protein expression data predict which fluxes, which so far is considered to explain only on the order of 1/3 to 1/2 of the fluxes in metabolism. Some seemingly moderate adjustments are needed for the metabolic correlation analysis. A rather broad sweeping claim is made for three "paradigms" of enzyme levels being associated with fluxes through the immediate reaction, the pathway, and some other coupled reactions that are further away. Making several simplifications and strong assumptions by replacing essentially every data type used for yeast with something else, the MS ends with a short section on trying to make a similar case for human metabolism. Overall, I miss biological insights/novelty and remain skeptical about the relevance of the 3 paradigms. Some observations correlate but the advance over previous work is not clear, causality is not established, and there is no attempt to actually test the conclusions, which in some cases are even wrong as outlined below.

We thank the reviewer for the critical comments. We agree with the concern regarding the lack of causality in our initial conclusions on flux regulation mechanisms. While our intention was to understand flux regulation through a systematic analysis of the correlations between flux and enzyme levels, we acknowledge the limitations imposed by insufficient data and the issues inherent in correlational studies. In response, we have carefully revised the manuscript to avoid overinterpretations, such as the three paradigms. We also clarified that the study's objective is to understand the association between changes in flux and gene expression. Our revised manuscript now emphasizes the eFPA method, which leverages findings of local-pathway associations between enzyme levels and flux to predict changes in flux based on enzyme expression.

We believe the revised manuscript clarifies the novelty and scope of our study. The implications for flux regulation are now confined to the *Discussion* section, which should sufficiently address the technical concerns. Although we have shifted the manuscript's focus from flux regulation to method development, the core achievements remain the same. Our work represents a significant advancement in understanding the physiological relevance of metabolic gene expression regulation (i.e., how these regulations connect to flux) and establishes a robust approach for analyzing metabolic changes using gene expression data.

Other concerns are addressed in the following responses to specific points.

Main points:

1. The 3 paradigms are not new. Nobody doubts that enzyme levels influence the catalyzed reaction, the pathway flux and in some cases more distant fluxes (ie flux coupling), the latter being a well-known and integral element of flux balance analysis. Previous work did find some but not very strong correlation of enzyme/mRNA levels with catalyzed reactions and the local pathway flux. Although no quantitative comparison is offered in the present MS, it seems that somewhat more enzyme-flux correlations are found. My concern is that the reported results are more or less the same as in all previous studies in that enzyme levels correlate with/explain some fluxes to some extent. For yeast it is well known that for example respiration and TCA cycle are mainly transcriptionally controlled, presumably because of the major protein investment.

We agree that some aspects of the three paradigms are not new, and we have built our methods based on these. Our initial intention was to emphasize that, despite limited correlation between enzyme and flux at the reaction level (as many previous studies have found), flux changes are largely associated with coordinated enzyme level regulations in broad-sense pathways. These pathways are defined by network topology and flux balance, ranging from local classic pathways to globally coupled sub-networks. This latter discovery is novel, and leads to a significant increase in observed correlations, from 46 reactions at the reaction level to 73 in local-pathway contexts and 101 in optimal-boundary pathway scenarios, as illustrated in Fig. 2h. These findings have also been verified by statistical analyses (Fig. 2i, EV4a). Furthermore, our data interpretation method that integrates gene expression in local pathways (local-pathway eFPA) is now shown to be a better predictor than not only individual reaction expression but also other integrative methods that use gene/enzyme expression to infer flux changes (Figs. 2f, 2j, 5, EV5).

We realize, however, that our original presentation might have led to confusion due to the introduction of new terminologies and concepts. To address this, we have streamlined the manuscript to focus on the core discovery of local-pathway associations and removed potentially confusing terms like "enzyme reach" and the "three paradigms". The revision should more clearly communicate the novelty of our study.

The main difference of the present work is that the authors include, beyond central metabolism, also about 100 biosynthetic reactions for which fluxes can be easily inferred from biomass formation. These biosynthesis reactions are the simpler part that again is well known to be under strong genetic control, in particular in the used data set where amino acids were used for perturbation and where the increasing growth rate in chemostats depends in a linear fashion on higher flux through biosynthesis reactions. There is a danger that the chosen data sets and the focus on biosynthesis reactions (about 2/3 of all in the chosen network and manually defining 20 synthesis pathways for pathway correlation; line 9 p 7) introduces a strong bias. Along these lines I believe it is a misconception that "enzyme usage significantly impacts flux in amino acid biosynthesis" with reference to other work (P 6). If more biosynthesis is necessary biosynthesis genes are expressed stronger - which still does not mean that they control or cause the flux through a given reaction. Some correlation with the biosynthesis flux is of course expected and known, but it does not explain the flux in any quantitative sense.

We appreciate the reviewer's insights regarding the correlation-causation issue. We agreed that it is overly simplistic to infer mechanistic regulation of flux solely from correlation analyses. Accordingly, we have adjusted our manuscript to focus solely on exploring associations between flux and enzyme levels rather than drawing definitive conclusions about flux regulation mechanisms.

However, we respectfully disagree with the concerns regarding potential biases in our dataset selection. We believe that the SIMMER dataset used here represents one of the most comprehensive and systematic dual omics datasets available. This dataset covers more than 200 reaction fluxes involving both catabolism and anabolism. It is not confined to amino acid biosynthesis; it also includes extensive data on nucleic acids, lipids, and polysaccharides metabolism. Contrary to the suggestion that these fluxes are "easy" to infer, accurate determination requires meticulous measurement of numerous biomass precursor effluxes in chemostats to meaningfully constrain the solution space in flux variability analysis (with less than 30% uncertainty as detailed in Hackett et al., Science, 2016). The comprehensiveness of the SIMMER dataset distinguishes it from other available datasets (detailed in Table EV1), which informed our decision to focus on analyzing this dataset.

Additionally, we have implemented rigorous normalization procedures to mitigate potential confounding effects associated with growth rate scaling in chemostat cultures. Specifically, all fluxes were normalized by dividing by the growth rate of the corresponding condition, converting them into relative fluxes. These were then compared with relative enzyme levels measured by proteomics (see Appendix for further details). This normalization process ensures that observed correlations are not merely reflections of passive scaling with growth rate.

Overall, it remains unclear what the really novel contribution of this work is and how much better it explains what?

We believe that, having addressed these and other concerns and focused more on method development for data integration, the novelty of the work and improvement against existing methods/knowledge should be now clear in the revised manuscript.

2. The enzyme reach aspect seems to be primarily a reinvention of the well-known flux coupling. Before it can be claimed to be a "novel" parameter, it would need to be formally checked against the general flux coupling that is inherent to metabolic networks. My expectation is that enzyme reach is only a subset of generally flux coupled reactions. The claim that this work "revealed" that enzymes can influence also distant reactions (eg start of discussion and elsewhere) is not justified by any means. This has long been known.

We appreciate the reviewer's critical perspective on the concept of "enzyme reach" and the comparison to flux coupling. Acknowledging the potential for confusion with existing concepts, we have eliminated the term "enzyme reach" from our manuscript and used direct descriptions when applicable. However, we would like to clarify that the relationship between target reaction (i.e., reaction of interest, or ROI) and local pathway reactions used by eFPA is different from flux coupling. We agree that it has long been recognized that enzymes can influence fluxes beyond their immediate reactions due to flux coupling, or mass balance in steady states. What remains less understood, and what our study contributes, is the empirical relationship between enzyme levels and flux changes, i.e. which enzymes in the network are more relevant to a flux of interest. Our systematic analysis reveals that significant associations are predominantly localized within local pathways defined not by arbitrary classifications but by actual metabolic and topological constraints. Relevant local pathways are generally within six steps of network

distance and are characterized by flux-balanced routes through metabolites with low network degrees. Such local-pathway associations, initially referred to as “enzyme reach”, are implemented in eFPA through two novel parameters, weighted metabolic distance and distance boundary. To better illustrate this point, we have included clarifications in the revised *introduction* (page 2 line 33 to page 3 line 27).

On the other hand, we do agree with the reviewer that the analysis regarding distal enzyme reach mediated by flux coupling (original Fig. 3) was rather confusing and hard to justify. Therefore, we removed the corresponding figure from our new submission to maintain focus on the validated core discovery of local pathway association and the methodological improvement based on this finding.

3. By remaining strictly at the level of correlations, the work sidesteps the more relevant causality. To make the work meaningful it would be important to explain how the newly found correlations help to overcome the general consensus in the field that fluxes are NOT controlled by expression, which is based on several key observations such as i) enzymes operate typically operate well below their in vitro K_m and V_{max} ; ii) enzyme overexpression typically does not increase flux (neither in the reaction nor the pathway); iii) modulating enzyme levels up/down over quite a range does not affect flux; iv) the timing of expression does not match most metabolic responses in microbes such as yeast; and v) expression changes were repeatedly shown to be poor predictors of what cells need in a given conditions (Price, Deutschbauer et al 2013 MSG or Birrell, Brown et al 2002 PNAS and many others).

We thank the reviewer for bringing up this long-standing paradox in the field: experimental analyses often show little correlation between flux and enzyme levels for most reactions, yet numerous metabolic network modeling studies assuming some level of correlation have successfully derived predictions about flux from gene/protein expression. This discrepancy may be caused by a missing component in experimental studies: the integration with the metabolic network. Our study explored this gap by systematically analyzing the experimental data for correlation between flux changes and enzyme levels across different scales of integrated network regions. We discovered that flux changes correlate more strongly with coordinated changes in enzyme expression within local pathways. This finding challenges and potentially expands upon the five observations mentioned by the reviewer. For instance, with regard to the observation that expression changes are poor predictors of cellular needs (observation #5), our study suggest that previous approaches might have overlooked the importance of network integration in translating expression changes to cellular needs. Our eFPA algorithm may facilitate a reevaluation of these data focusing on coordinated changes in local pathways, rather than expression changes at reaction level, in future studies.

To clarify the relevance of our findings in this context, we now include a discussion of this paradox in the *introduction* (page 2 line 23-32). Furthermore, from a practical point of view, the existence of correlation between enzyme levels and flux is sufficient for data integration, no matter if there is a direct causality relationship or not. Thus, in our revised manuscript, this paradox has become a side issue, as we have indicated in the Discussion section (page 14, line 24-33).

Moreover, alluding to MCA in the discussion that this theoretical concept of shared flux control would be supported by the present analysis is a misconception. MCA is a great theoretical framework but works only for infinitesimally small changes in expression upon which the control would immediately shift to another reaction - not for the huge steady state values considered here.

We respectfully disagree. While it is true that MCA is mathematically derived in a differential equation form that corresponds to infinitesimally small changes in enzyme levels and its effect on flux (like a slope of a curve), this theoretical foundation does not preclude it from being tested and conceptualized in the integral form. Historically, classic experimental validations of MCA have indeed involved titrating enzyme levels to determine control coefficients and have compared the effects of manipulating single versus multiple enzymes in a pathway (e.g., PMID: 1445205). Our study's use of steady-state data and correlation analysis aligns with these principles and illustrates that flux regulation in vivo often involves coordinated control by multiple enzymes. Therefore, we believe that maintaining the conceptual link to MCA in the discussion is both appropriate and informative, which helps bridge theoretical frameworks with empirical observations. In the context of the revised manuscript, we bring up this idea as a discussion point with regard to flux control, which is toned down compared to the original text (page 14, line 18-21).

4. There are only very few biological interpretation attempts of observed correlations and in some cases they are wrong. On page 10 the correlation of fluxes to glutathione with those through glycolysis, ox phos and TCA cycle is claimed to suggest that glutathione synthesis would be energy limited. Firstly, there is no reason whatsoever why the synthesis of this tripeptide should be energy-limited. Second, the situation is more likely be the reverse. Increased respiration generates more ROS, which in turn requires more defense molecules such as glutathione - a long and well-known relationship. Besides the obviously wrong interpretation, this illustrates the problem with having only correlations.

This comment is less relevant to the revised manuscript as we changed it to a method-oriented paper and the pertaining figure was removed. However, we respectfully disagree with the reviewer regarding the glutathione example. In the original manuscript, we referred to an independent experimental study that demonstrated glutathione synthesis in yeast is energy limited, to support our predictions derived from correlation. We therefore believe our original claim was well justified by the experimental evidence although it may not be intuitive.

5. The final section on human metabolism is only possible by making sweeping assumptions such as replacing protein with mRNA data and reaction fluxes with uptake rates. These are completely novel input data that require lots of (also partly questionable) changes and assumptions that are only dealt with as an afterthought. The claim that mRNA levels can be reliably used instead of proteomics goes against a huge body of evidence suggesting that the two quantities are not well correlated. How is it then possible that it works just as well in eFPA and what does it mean for that analysis? In general, this section is not well supported beyond a superficial analysis and is probably better removed.

We respectfully disagree with these critiques. First, the approach of using boundary flux predictions by eFPA was initially validated using yeast data, as shown in Fig. EV4b,c, before extending this methodology to human metabolism. This progression demonstrates a methodological consistency rather than an ungrounded assumption. Second, regarding the correlation between mRNA and protein levels, it is important to clarify that our analysis does not overlook the well-documented disparities between these two quantities in terms of their absolute levels, e.g., TPM of mRNA and copy numbers of protein. Instead, our focus is on the relative changes across conditions, which can exhibit significantly different correlation dynamics. Indeed, relative levels across conditions often show a stronger correlation, particularly for metabolic genes, as evidenced by previous studies (e.g., PMID: 24762675). In addition, the correlation between mRNA and protein levels is context-dependent, with certain conditions and gene

groups better correlated than others. For instance, the relative levels of protein and mRNA across 32 human tissues demonstrate a notable correlation for most genes, including metabolic genes, as reported in Jiang et al. (PMID: 32916130). We also show the distribution of correlation coefficients in the figure below. Notably, the eFPA integration further increases the correlation, which is possibly because of the smoothing effect when combining information in local pathways.

Furthermore, we would like to emphasize that mRNA data is routinely used as a proxy for enzyme levels in many studies aiming to infer metabolic network functions across various tissues and conditions (please see Introduction page 3 line 3-13, and references therein; and further discussion on page 15 line 5-10). Ideally, eFPA would be recalibrated for mRNA data given its initial optimization with proteomic data, but the lack of a matching mRNA dataset that fulfills our quality criteria prevents such recalibration at the present. Nevertheless, our analyses demonstrate that eFPA performs comparably well with both proteomic and transcriptomic data in human tissues. Additionally, it shows improved predictive accuracy over alternative methods when applied to single-cell data. Please see section ***eFPA enables metabolic analysis at single-cell level*** from page 9, line 34 to page 14, line 5.

The test itself in this section is also not clear. What does it mean that "... we could validate boundary flux predictions with available tissue-enriched metabolite annotations" (P 13 L 12/13)? An annotation is only the name of a metabolite? It is also a major restriction to have only uptake and not intracellular fluxes, all of which is not well explained and discussed in terms of consequences.

We appreciate the reviewer's feedback on the clarity of our testing methods. In response, we have revised the relevant section (page 8 line 21-31) to better explain the approach. The term "annotation" in the following paragraph (page 9, line 6) refers to a list of metabolites that are enriched in specific tissues, covering 17 of the 32 tissues analyzed.

We also appreciate the question about direct validations with intracellular fluxes. However, such data is not available at scale for a systems-level validation. To partially validate our method, we performed additional literature review and identified several isotope tracing studies in mouse that provides measurements of flux contribution to tissue TCA cycle from different metabolites. Importantly, we found that the mouse tissue data supports our prediction about the fatty acid oxidation flux in the heart, which we originally used as an example for predictions that are dependent on eFPA. This additional validation is now included in in Fig. 3d.

Minor points

1. Text contains many generalisms and unclarities. Here are a few examples:
- p 2 bottom: the artificial separation of expression mechanisms in innate (during development) and in response to conditions. Even developmental changes are based on exogenous factors, besides it would no sense for the here studies yeast anyway. From this rewiring of networks is separated from wiring. This is not substantiated by any evidence.

We agree with the reviewer and have removed these statements.

- unclear what "flux data adjusted to growth rates" are. The usual concept is to normalize to the uptake rates so that relative fluxes are obtained.

Thank you for raising this point regarding the clarification needed on flux data normalization. In response, we changed "adjusted" to "divided by" for clarity (page 4 line 33). More explanations for this normalization are included in the responses above and detailed in Appendix (page 2 Equation S2 and corresponding texts).

We acknowledge that normalizing by the uptake rates is an alternative approach, but we argue that normalizing by a single uptake reaction rate is not ideal because different conditions have varying active uptake reactions. On the other hand, the absolute fluxes in a chemostat setting are well-known to consistently scale with the growth (dilution) rate. Therefore, it is both appropriate and necessary to obtain relative flux by growth rate normalization.

2. On P 9 the reaction network has 232 reactions while on P 5 it is 156?

We have explained in the main text (page 4 line 23-27) that 232 reactions refer to all reactions with valid flux measurements in SIMMER dataset and 156 refers to all reactions with both valid flux and protein level measurements.

Reviewer

#2:

In the manuscript, "Linking enzyme expression to metabolic flux," Dr. Walhout and colleagues examine the extent to which enzyme expression levels are correlated to flux through metabolic pathways when network architecture is accounted for. Though previous studies have shown that metabolite availability predicts reaction fluxes better than enzyme availability, the authors note that previous work has emphasized individual reaction fluxes, rather than taking a network approach. Using a previously published data set of reaction fluxes and enzyme levels in yeast under various environmental conditions, they find that if enzymes regulating closely related pathways are co-expressed, there is a stronger correlation between enzyme levels and flux. This result suggests that enzymes have a "reach" of influence within a network, which extends

beyond the specific reactions they catalyze. The authors incorporated this concept of enzyme reach into a previously published algorithm (FPA), and applied this modified algorithm (eFPA) to predict flux values from enzyme expression levels in both yeast and human tissues.

We thank the reviewer for the nice summary of our study.

Critiques

The central concept of the paper is interesting, but the paper lacks a clear rationale...what does the result imply about the coupling or regulation of enzymatic flux? How could approaches for integrating expression data into network models be improved given the results presented?

Thank you for your feedback. We appreciate the reviewer's recognition of our work and agree with comments regarding the initially unclear rationale and implications for future method development. This should now be clear as we have shifted the emphasis toward method development and have shown the improvement in predictive power over alternative approaches. The addition of our new section on the application of eFPA to scRNA-seq data directly addresses the reviewer's comment, further illustrating the practical utility and enhanced predictive capability of our approach.

If we assume that the results of the paper allow for better integration of expression data into network models (e.g., enzyme expression that correlates with flux could be handled differently than enzymes that don't correlate with flux), what kind of concerns are there with the fact that the analyses are based on a single data set? How generalizable are the results so that those concepts could be applied to another system for which there isn't coupled expression and flux data?

Thank you for the constructive question. We now include the discussions of the potential limitations related to single-dataset analysis in the *Discussion* section (page 15 line 1-13). We clarified that while the associations between flux and coordinated regulation of enzyme levels in local pathways are likely to be generally applicable, our method was optimized using the yeast proteomic data.

To further substantiate the general applicability of our method, we had already conducted validations that show consistent results using both transcriptomic and proteomic data from human samples in the original manuscript. Moreover, our new study on single-cell data demonstrates that our methodological choice of a distance boundary of 6 for local pathway analysis is appropriately conservative, supporting the broad application of the eFPA approach. Additionally, the benchmarking of our method against both yeast and human single-cell data, included in the revised manuscript, addresses the concern about generalizability. These comparative analyses show that our method can effectively integrate expression data into network models across different biological systems and conditions.

The number of new concepts (enzyme reach, eFPA, etc.) makes the paper very difficult to understand. Similarly, there are so many "co" words (correlated, collinear, coupled, co-expression, connectivity); it's hard to keep track of what they mean in the context of this specific study.

We agree with the reviewer's suggestion and have avoided introducing unnecessary new terminology in this submission. Additionally, we have removed the original Fig. 3, which involved

collinearity and coupling analysis, to focus solely on the major discovery, as detailed in the response to reviewer #1 (major point #2).

Relatedly, the figures are extensive and detailed, but at the expense of interpretability. Text is very tiny (especially EEV4 and EV5) and there's just so much data and analyses in each of the 4 main figures, each of the 5 extended figures, and all the supplement. It seems the work would be better suited split up into a couple papers or streamlined even more so that the essence of the messages aren't lost.

We thank the reviewer for acknowledging our extensive efforts. To enhance readability for publication, we streamlined the manuscript by removing the original Fig. 3 and focusing solely on the core discovery of local-pathway association. Although we introduced a new section on single-cell analysis, we believe that the revised submission maintains a more linear logical flow, from development to application of eFPA, and is easily comprehensible.

In human tissues, how do eFPA predictions of metabolite fluxes from mRNA compare to other methods of predicting fluxes that don't include enzyme reach? Are there key differences or additional insights that this method offers?

We addressed this question in the original manuscript by comparing eFPA results with the expression of transport reactions. This comparison demonstrates that eFPA, by integrating local enzyme activities, provides a more comprehensive analysis than focusing solely on metabolite transport. For benchmarking in the revised manuscript, we compared eFPA with other integrative methods for yeast and single-cell data. However, we did not extend detailed benchmarking to metabolite analysis because other methods are not typically designed for metabolite-level analysis. Also, the primary aim of this section in the revised manuscript, along with other analyses of human bulk data, is to demonstrate that both mRNA and proteomic data can be effectively integrated using eFPA, yielding consistent and comparable results.

Page 3, lines 19-23 seem to conflict with the assertion made on page 6, lines 22-23

Thank you for pointing this out. We acknowledge that the original phrasing was confusing. We intended to indicate that the regulation of flux at single reaction level in a few amino acid biosynthesis pathways is well-documented, rather than that across most reactions in the network. We have revised the sentence to clarify this distinction (page 5 line 10-18).

Page 8 - briefly explain the concept of metabolite degree. For example, "Metabolite degree captures the total number of sources and sinks for the metabolite..."

Thank you and we revised the sentence (page 6 line 15).

Figure 2A - typo ("consumption")

Thank you.

Figure 3a - don't understand the difference between the black line and blue line

This figure has been removed.

Reviewer #3:

In this study the authors aim to get insight into how flux is controlled in metabolic networks. The authors state that previous studies demonstrated that flux is controlled by metabolite concentration rather than by enzyme concentration, and here the authors proposed the control of enzymes only on fluxes of non-cognate reactions. The authors use data from yeast in their study and in the last part of the paper they extrapolate their findings from yeast also to human metabolism. Overall the finding in this study is interesting but there are quite a few terminologies not appropriately explained in the manuscript, preventing full understanding and assessment of the content. Please consider the comments below.

Thank you for your appreciation of our study. We considered all comments to improve the manuscript and made major adjustments as outlined above. We hope the contents of the new manuscript are clear.

What is the biological basis of the hypothesis that enzyme levels affect metabolic flux of non-cognate reactions? In other words, if a good correlation is observed between an enzyme and flux of a non-cognate reaction, by which biological mechanism the enzyme controls the non-cognate reaction?

Thank you for raising this question. While the biological mechanism of indirect controls is beyond the scope of this study, it has been addressed in Metabolic Control Analysis (MCA) and subsequent studies (e.g., PMID: 1530563). As also noted by reviewer #1, metabolic fluxes are interdependent due to mass balance at steady state, meaning the regulation of one reaction flux impacts other connected reactions in the network. We have included this explanation in both the introduction (page 3 line 1-3) and discussion (page 14 line 16-23) in the revised manuscript.

New terminologies should be clearly defined, especially cognate reaction and enzyme reach.

Thank you and we removed the confusing terminologies. The term "cognate" appears only once in the discussion in a context where its meaning can be inferred from the surrounding text (page 14 line 17-18).

It would be better to demonstrate whether and how much eFPA outperforms the original FPA.

Thank you and we added the benchmark analysis in Fig. 2j (page 8 line 9-18) and Appendix Fig. S2b (as referenced on page 7, line 22-24).

P8 L20-23: does FPA calculate relative flux potential (rFP) or flux potential (FP)? What is the output of FPA?

Thank you for pointing out the unclear sentence. FPA calculates the flux potential (FP), which is then converted to the relative flux potential (rFP) by dividing FP by its theoretically maximum. We have clarified this process in the revised text (page 6 line 32 to page 7 line 1).

P9 L14: what is "the corresponding eFPA"? Is eFPA here the algorithm itself or an output (e.g. maybe rFP or FP) of this algorithm?

Thank you for pointing out the unclear sentence. The term "corresponding eFPA" refers to eFPA with a set parameter (distance boundary = best-predicting distance of a target reaction). To

avoid confusion, we have changed the phrase to “the corresponding predictions as the results of optimal-boundary eFPA” in the text (page 7 line 13-15).

P10 L7: Fig. 2g appears to indicate that levels of metabolic enzymes do not affect their cognate reactions.

We revised the sentence to both clarify the expression and avoid making affirmative conclusions on flux regulation (page 7 line 32 to page 8 line 2). This figure primarily demonstrates that changes in the flux of a reaction of interest (ROI) are often predictable based on the expression changes of enzymes in nearby reactions within the same local pathway, typically due to co-expression of enzymes within that pathway.

Fig. 1e: in the upper box all the five enzymes are linked together but in fact they should be just pairwise linked. Is this also the same issue in Fig. 1f?

We agree with the reviewer that the comparison (PCC calculation) is pairwise, which is more aligned with a pairwise linkage in the cartoon. However, due to space limitations, visualizing the pairwise linkage is challenging. Therefore, we decided to retain the current visualization.

Legend of Fig. 3a: "The number of reaction fluxes predicted by either collinearity or eFPA are shown in red". 1) Collinearity is defined as PCC between two fluxes (P11 L9) but how does it predict fluxes? 2) While it is not clear eFPA predicts rFP or FP, both rFP and FP are different from reaction flux.

We acknowledge the reviewer's question regarding the use of collinearity as a general predictor of flux. However, this question is not relevant anymore as we have removed this figure.

Fig. 3bc: what is the relative flux? Is it rFP?

This figure was removed during our revision.

5th Nov 2024

Manuscript Number: MSB-2023-11970R-Q

Title: Enhanced flux potential analysis links changes in enzyme expression to metabolic flux

Author: Xuhang Li

Albertha Walhout

Lutfu Yilmaz

Dear Dr. Yilmaz,

Thank you for submitting your work to Molecular Systems Biology. We have now heard back from the three original referees who agreed to evaluate your manuscript. As you will see below, although Reviewers 2 and 3 are satisfied with the revised version of the manuscript, Reviewer 1 continues to have substantial concerns, particularly on overstatements, unclear methodological advance (but lacking full expertise to evaluate this), and limited biological insights. Based on this, and the fact that the paper was refocused from the previous version from an article to a computational method, we sought the advice of an additional expert. Briefly, this advisor mentioned the following:

- "the manuscript introduces new ill-/non-defined terms/language/concepts ("local-pathway integration", "localized, relevant pathways", etc.)", which make it difficult to understand the work, which was also brought up by multiple reviewers in the first round of review.
- "it is not fully clear what the main contribution of the work is: The abstract mentions the problem that a systematic investigation of how enzyme expression changes relate to flux has been lacking. But at the same time the abstract indicates that the authors rather propose a method, i.e. 'enhanced flux potential analysis' (eFPA) algorithm, which is what apparently the authors focus on as stated in the response letter. If it is the method, then: Flux Potential Analysis (FPA) has been previously published. It is questionable whether 'enhanced flux potential analysis' (eFPA) algorithm represents the advance is sufficient to warrant publication in MSB."
- "...compared to the previous manuscript, the author removed claims on biological insights. Thus, what is then left is the use of empirical relationship between enzyme levels and flux changes for metabolic flux predictions. It is somewhat unsatisfactory to employ such relationships in flux predictions without having a solid mechanistic basis for those relationships."

Therefore, we would ask you to address the reviewer concerns in a revision focused on additional benchmarking of eFPA compared to FPA to clarify the advance of the method, clearly define any new concepts and tone down overstatements, and finally address whether/why it is justified to use the empirical relationships between enzyme levels and flux changes in your paper.

If you feel you can satisfactorily address these points and those listed by the referees, you may wish to submit a revised version of your manuscript with a point-by-point response to the three reviewers and points raised by the expert as mentioned above. If you would like to discuss further the points raised by the referees, I am available to do so via email or video. Let me know if you are interested in this option.

We require:

- 1) A .docx formatted version of the manuscript text (including legends for main figures, EV figures and tables). Please make sure that the changes are highlighted to be clearly visible. Alternatively you may choose to submit your manuscript as a LaTeX file.
- 2) Individual production quality figure files as .eps, .tif, .jpg (one file per figure). For guidance, download the 'Figure Guide PDF' (<https://www.embopress.org/page/journal/17574684/authorguide#figureformat>).
- 3) At EMBO Press we ask authors to provide source data for the main figures. Our source data coordinator will contact you to discuss which figure panels we would need source data for and will also provide you with helpful tips on how to upload and organize the files.
- 4) A .docx formatted letter INCLUDING the reviewers' reports and your detailed point-by-point responses to their comments. As part of the EMBO Press transparent editorial process, the point-by-point response is part of the Peer Review File (PRF), which will be published alongside your paper.
- 5) A complete author checklist, which you can download from our author guidelines (<https://www.embopress.org/page/journal/17574684/authorguide#submissionofrevisions>). Please insert information in the

checklist that is also reflected in the manuscript. The completed author checklist will also be part of the PRF.

6) Please note that all corresponding authors are required to supply an ORCID ID for their name upon submission of a revised manuscript.

7) It is mandatory to include a 'Data Availability' section after the Materials and Methods. Before submitting your revision, primary datasets produced in this study need to be deposited in an appropriate public database, and the accession numbers and database listed under 'Data Availability'. Please remember to provide a reviewer password if the datasets are not yet public (see <https://www.embopress.org/page/journal/17574684/authorguide#dataavailability>).

In case you have no data that requires deposition in a public database, please state so in this section. Note that the Data Availability Section is restricted to new primary data that are part of this study. This study includes no data deposited in external repositories.

8) All Materials and Methods need to be described in the main text using our 'Structured Methods' format, which is required for all research articles. According to this format, the Methods section includes a Reagents and Tools Table (listing key reagents, experimental models, software and relevant equipment and including their sources and relevant identifiers) followed by a Methods and Protocols section describing the methods using a step-by-step protocol format. The aim is to facilitate adoption of the methodologies across labs. Please upload the Reagents and Tools table as a separate document when submitting your revised manuscript. More information on how to adhere to this format as well as a downloadable template (.docx) for the Reagents and Tools Table can be found in our author guidelines:

<https://www.embopress.org/page/journal/17444292/authorguide#structuredmethods>

9) For data quantification: please specify the name of the statistical test used to generate error bars and P values, the number (n) of independent experiments (specify technical or biological replicates) underlying each data point and the test used to calculate p-values in each figure legend. The figure legends should contain a basic description of n, P and the test applied. Graphs must include a description of the bars and the error bars (s.d., s.e.m.). Please provide exact p values.

10) Our journal encourages inclusion of *data citations in the reference list* to directly cite datasets that were re-used and obtained from public databases. Data citations in the article text are distinct from normal bibliographical citations and should directly link to the database records from which the data can be accessed. In the main text, data citations are formatted as follows: "Data ref: Smith et al, 2001" or "Data ref: NCBI Sequence Read Archive PRJNA342805, 2017". In the Reference list, data citations must be labeled with "[DATASET]". A data reference must provide the database name, accession number/identifiers and a resolvable link to the landing page from which the data can be accessed at the end of the reference. Further instructions are available at .

11) We replaced Supplementary Information with Expanded View (EV) Figures and Tables that are collapsible/expandable online. A maximum of 5 EV Figures can be typeset. EV Figures should be cited as 'Figure EV1, Figure EV2" etc... in the text and their respective legends should be included in the main text after the legends of regular figures.

<https://www.embopress.org/page/journal/17574684/authorguide#expandedview>

13) Author contributions: CRediT has replaced the traditional author contributions section because it offers a systematic machine readable author contributions format that allows for more effective research assessment. Please remove the Authors Contributions from the manuscript and use the free text boxes beneath each contributing author's name in our system to add specific details on the author's contribution. More information is available in our guide to authors.

Please also suggest a striking image or visual abstract to illustrate your article as a PNG file 550 px wide x 300-600 px high. Share synopsis text and image, as well as eTOC:

Please note that these would be the final versions and changes during proofing are usually not allowed

16) As part of the EMBO Publications transparent editorial process initiative (see our policy here: https://www.embopress.org/transparent-process#Review_Process), Molecular Systems Biology will publish online a Peer Review File (PRF) to accompany accepted manuscripts.

In the event of acceptance, this file will be published in conjunction with your paper and will include the anonymous referee reports, your point-by-point response and all pertinent correspondence relating to the manuscript. Let us know whether you agree with the publication of the PRF and as here, if you want to remove or not any figures from it prior to publication.

Please note that the Authors checklist will be published at the end of the PRF.

Molecular Systems Biology has a "scooping protection" policy, whereby similar findings that are published by others during review or revision are not a criterion for rejection. Should you decide to submit a revised version, I do ask that you get in touch after three months if you have not completed it, to update us on the status.

I look forward to receiving your revised manuscript.

Yours sincerely,

Poonam Bheda

Poonam Bheda, PhD
Scientific Editor
Molecular Systems Biology

Reviewer #1:

The revised version is essentially a new manuscript. What used to be a manuscript investigating yeast flux regulation with a variant of FBA that claimed three paradigms of enzyme abundance-flux relationships is now turned into a methodological flux estimation paper that is "calibrated" on yeast data and then mainly applied to human data. For this reviewer it is more like a new submission that should primarily be evaluated by its computational methods merits as there are no or only very limited biological insights.

1. I maintain my initial concern that the underlying logic of the presented computational method is neither new nor surprising, i.e., that concerted changes in pathway gene expression are indicative of changes in pathway fluxes. There is broad consensus in the field that considering expression changes in their biological context is more meaningful than considering them individually, and this principle has been exploited in prior flux studies. I therefore disagree with the somewhat pompous claim in the introduction that the study addresses "... a fundamental question about how changes in enzyme expression translate to changes in flux." We know very well the various factors that influence flux, we just don't know which of them matters to which extent under what condition. The presented correlative approach cannot address this question from any causal level.

Instead, the relevant contribution of this work is solely a methodological one in developing a computational method that claims to be better than existing ones in predicting fluxes in an FBA framework by exploiting expression data. If the advance is significant, that would certainly be relevant. Hence the merit of this work should primarily be evaluated on the basis of this improvement. The main evidence for that is a benchmarking in Fig. 2j showing a higher number of reactions where predicted fluxes consistently correlate with experimentally estimated fluxes. This is a good start, but since this is the main evidence presented, one would have liked to see a deeper treatment of the matter like how good these predictions are, why the other two thirds of fluxes are not correlating, or what we can learn from what we can and cannot predict from expression data. In particular I would be interested to see whether the additionally correctly predicted reaction fluxes are more trivial cases that arise from using more information in eFPA or truly hard to predict cases with several degrees of freedom.

With no criticism to the authors, this reviewer does not feel competent to judge the new computational algorithm and its novelty and advance over existing ones. This should be done by an algorithm developer, for example one of those who developed the benchmarked methods.

2. The initially very small section on application to human metabolism has now become the major part of the manuscript. The biological insights are limited and restricted to comparative statements on some fluxes where prior methods led to different results. Methodologically, there seem to be some useful adjustments to prior methods in how to deal with notorious missing values in single cell data that could be useful for single cell analyses.

I struggle with the key adjustment that was necessary to move from yeast (where experimentally estimated flux data was available) to human tissues where that is not the case. Instead of using the hard to estimate intracellular fluxes, now the authors (have to) use the much easier to estimate exchange fluxes with the environment as a postulated proxy. Nobody would doubt that higher expression of proteins in a degradation/synthesis pathway would at least be indicative of such cells taking up and degrading/secreting the cognate metabolite from the environment, say an amino acid that can be used as an energy source, and this concept has been used multiple times. Again, I cannot judge from the presented information how novel and how much better the proposed eFPA is.

Reviewer #2:

I still think there is too much data/analysis for one manuscript like this, but I think the authors sufficiently addresses the critiques and it will be nice for the readership to evaluate the work.

Reviewer #3:

I think the authors have addressed all critical questions well and the revised version of the paper is much improved. I have no further comments.

Response:

We thank the editor, the reviewers, and the advisor for their insightful feedback. We have addressed all comments by the reviewers and the advisor and have revised the manuscript in accordance with these comments and the editor's guidelines. We believe this revision refined our manuscript into a more robust methodological contribution that fits well with the interests of the *Molecular Systems Biology* readership.

In the revised manuscript, we have clarified the innovations and advantages of the enhanced flux potential analysis (eFPA) over its predecessor. We have simplified or explicitly defined previously unclear terms and adjusted the narrative to better emphasize our objective: developing methods for systematic interpretation of gene or protein expression changes as they relate to metabolism.

Detailed, point-by-point responses to each comment are highlighted in blue text below.

Advisor:

1. "the manuscript introduces new ill-/non-defined terms/language/concepts ("local-pathway integration", "localized, relevant pathways", etc.)", which make it difficult to understand the work, which was also brought up by multiple reviewers in the first round of review.

We have simplified the terminology throughout the manuscript, avoiding redundant words like "local" and "localized" when pathways are mentioned. Additionally, we have provided explicit definitions for essential terms at their first mention in the manuscript, such as "integrated pathways".

2. - "it is not fully clear what the main contribution of the work is: The abstract mentions the problem that a systematic investigation of how enzyme expression changes relate to flux has been lacking. But at the same time the abstract indicates that the authors rather propose a method, i.e., 'enhanced flux potential analysis' (eFPA) algorithm, which is what apparently the authors focus on as stated in the response letter. If it is the method, then: Flux Potential Analysis (FPA) has been previously published. It is questionable whether 'enhanced flux potential analysis' (eFPA) algorithm represents the advance is sufficient to warrant publication in MSB."

We appreciate the feedback, which highlights the need for further clarifications of the main advances presented in our study.

In the revised manuscript, we hope we better convey that we have developed a robust method to systematically interpret metabolic gene/protein expression datasets to better predict metabolic network activity. Importantly, we show that our method can predict metabolic activity as relative flux potentials not only for reactions associated with enzymes that change in expression level, but also for reactions where enzymes are regulated by other mechanisms, are unregulated, or even undetected, as shown in multiple cases with both yeast proteomic and human single-cell RNAseq data.

While the original Flux Potential Analysis (FPA) provided a foundation for this study, FPA was an initial heuristic extension to a non-quantitative network-level integration algorithm (of the iMAT family) and was not optimized to quantitatively predict flux based on network-level data involving both gene expression and flux levels.

Our systematic analysis with yeast data has significantly improved FPA, introducing three key refinements: (i) validating the concept of integrating enzyme expression data at the pathway level

using actual flux measurements, (ii) adjusting the rules of the algorithm to better capture the relationship between expression and flux, and (iii) defining and optimizing the algorithm's parameters based on real data. These improvements position eFPA as an advanced, optimized algorithm that surpasses FPA and other available algorithms in inferring metabolic functions from gene/protein expression data. We have explained these improvements more explicitly in the revised introduction (please see Introduction, last two paragraphs).

The concerns raised by the reviewer about the predictive power of eFPA compared to FPA made us realize that the analyses in the original manuscript were not sufficiently clear or visible. We have addressed this by revising Figure S2 from the Appendix, which compares predictions from the yeast dataset using eFPA and FPA, into a new Figure 3b. This figure, combined with the previous Figure 2j (now Fig. 3a) and specific examples (Fig. 3c and d), now forms a comprehensive Figure 3 dedicated to benchmarking eFPA against other methods, with a particular focus on FPA. Additionally, we have conducted a comparative analysis of single-cell data using both FPA and eFPA, the results of which are presented in the new items Figure EV5 and Table EV9. This analysis shows that although FPA improves upon the analysis of individual reactions, eFPA offers superior predictive power, thus better accommodating the sparsity typical in single-cell data. We trust that these revisions clearly demonstrate the advancements of eFPA over FPA in the revised manuscript. Please also see our response to Reviewer 1, comment #2.

3. "...compared to the previous manuscript, the author removed claims on biological insights. Thus, what is then left is the use of empirical relationship between enzyme levels and flux changes for metabolic flux predictions. It is somewhat unsatisfactory to employ such relationships in flux predictions without having a solid mechanistic basis for those relationships."

We appreciate the reviewer's critical insights. While eFPA indeed employs a heuristic approach based on empirically established relationships, we respectfully disagree on several points. First, our work extends beyond merely proposing empirical relationships; it introduces a validated framework for inferring metabolic function from gene/protein expression data, with particularly novel applications to single-cell data analysis. Second, our algorithm has a mechanistic rationale: it assumes that enzyme expressions within the same pathway as the reaction of interest are significant, whereas those in more distant parts of the network diminish in relevance due to the dispersion of metabolic information. This approach is rooted in the understanding that variations in enzyme expression or flux in distant pathways may not directly impact the flux of a target reaction because their outputs can be utilized by multiple other pathways within the metabolic network. While we can outline such an overall mechanistic basis, the specific rules of flux-expression association and the relevance of distant pathways for all reactions of interest remain elusive, necessitating the use of heuristics and data-driven optimization. In the revised manuscript, we have included a new paragraph in the Discussion section to elaborate on these fundamental concepts (please see Discussion, third paragraph). Lastly, although employing a heuristic approach may not satisfy all critiques, the increased predictive power of eFPA compared to other methods demonstrates its practical efficacy and supports its utility in advancing our understanding of metabolic systems.

Reviewer #1:

The revised version is essentially a new manuscript. What used to be a manuscript investigating yeast flux regulation with a variant of FBA that claimed three paradigms of enzyme abundance-flux relationships is now turned into a methodological flux estimation paper that is "calibrated" on yeast data and then mainly applied to human data. For this reviewer it is more like a new

submission that should primarily be evaluated by its computational methods merits as there are no or only very limited biological insights.

1. I maintain my initial concern that the underlying logic of the presented computational method is neither new nor surprising, i.e., that concerted changes in pathway gene expression are indicative of changes in pathway fluxes. There is broad consensus in the field that considering expression changes in their biological context is more meaningful than considering them individually, and this principle has been exploited in prior flux studies. I therefore disagree with the somewhat pompous claim in the introduction that the study addresses "..... a fundamental question about how changes in enzyme expression translate to changes in flux." We know very well the various factors that influence flux, we just don't know which of them matters to which extent under what condition. The presented correlative approach cannot address this question from any causal level.

We appreciate the reviewer's comment and agree that the general underlying assumption of our algorithm is not novel and is consistent with the current knowledge. However, the strength and the extent of the association between enzyme expression and flux remained largely unquantified, and understanding this is essential to developing algorithms that reliably infer metabolic activity from expression data. This point aligns with the reviewer's next comment.

We also agree that there is not necessarily a causal relationship between flux and expression levels. This concern was addressed in the previous round, where we removed any implications of causality. However, we acknowledge that the sentence cited by the reviewer may still suggest causality, and we have therefore rephrased it to the following (Introduction, first paragraph; page 2, lines 32-34):

"This discrepancy suggests that the relationship between changes in enzyme expression and flux is not well defined, complicating the interpretation of expression data."

2. Instead, the relevant contribution of this work is solely a methodological one in developing a computational method that claims to be better than existing ones in predicting fluxes in an FBA framework by exploiting expression data. If the advance is significant, that would certainly be relevant. Hence the merit of this work should primarily be evaluated on the basis of this improvement. The main evidence for that is a benchmarking in Fig. 2j showing a higher number of reactions where predicted fluxes consistently correlate with experimentally estimated fluxes. This is a good start, but since this is the main evidence presented, one would have liked to see a deeper treatment of the matter like how good these predictions are, why the other two thirds of fluxes are not correlating, or what we can learn from what we can and cannot predict from expression data. In particular I would be interested to see whether the additionally correctly predicted reaction fluxes are more trivial cases that arise from using more information in eFPA or truly hard to predict cases with several degrees of freedom.

We are grateful for the constructive criticism. We would like to indicate that the original section on single-cell data is relevant to this comment, as it shows how eFPA can effectively predict metabolic functions in scenarios where traditional gene expression analysis fails. To more directly address the reviewer's comment, specific examples have now been included in the section discussing yeast, highlighting instances where eFPA successfully predicts reaction fluxes even in the absence of proteomic data, as well as cases where expression levels do not correlate with fluxes, likely due to other regulatory mechanisms (please see new figures 3c and 3d; relevant text in

Results, section “An enhanced Flux Potential Analysis algorithm that translates relative enzyme expression to relative flux”, last two paragraphs [page 8, line 24 to page 9, line 30]). Please also see our response to the Advisor, comment #2.

Regarding the 56% of reactions that do not show correlation (Figure 2i): since our analysis systematically tested the correlation of these fluxes with enzyme levels in nearby and distant pathways, and found no association, these reactions are likely regulated by other mechanisms such as metabolite concentrations, allostery, or others that do not have a causal or significant association with gene expression. The related statements are added to the Results section [page 8, lines 19-23]:

“The remaining 130 reactions likely represent cases where fluxes are not controlled or correlated with enzyme expression. Instead, these fluxes may be driven by metabolite levels or regulated through mechanisms such as allostery, which are not directly influenced by changes in enzyme expression”

With no criticism to the authors, this reviewer does not feel competent to judge the new computational algorithm and its novelty and advance over existing ones. This should be done by an algorithm developer, for example one of those who developed the benchmarked methods.

3. The initially very small section on application to human metabolism has now become the major part of the manuscript. The biological insights are limited and restricted to comparative statements on some fluxes where prior methods led to different results. Methodologically, there seem to be some useful adjustments to prior methods in how to deal with notorious missing values in single cell data that could be useful for single cell analyses.

We thank the reviewer for this comment. We agree that the eFPA algorithm is particularly effective in handling dropout events. Additionally, it demonstrates robustness in other scenarios where single reaction expression or network-level integration methods fail, as illustrated in Figures 5g-k and Figure 6.

4. I struggle with the key adjustment that was necessary to move from yeast (where experimentally estimated flux data was available) to human tissues where that is not the case. Instead of using the hard to estimate intracellular fluxes, now the authors (have to) use the much easier to estimate exchange fluxes with the environment as a postulated proxy. Nobody would doubt that higher expression of proteins in a degradation/synthesis pathway would at least be indicative of such cells taking up and degrading/secretory the cognate metabolite from the environment, say an amino acid that can be used as an energy source, and this concept has been used multiple times. Again, I cannot judge from the presented information how novel and how much better the proposed eFPA is.

We appreciate the reviewer's concern. In response to this and Reviewer 2's feedback on the extensive data analysis, we have removed the section on exchange flux potentials and metabolite enrichment to streamline our presentation and focus on core findings.

Reviewer #2:

I still think there is too much data/analysis for one manuscript like this, but I think the authors sufficiently addresses the critiques and it will be nice for the readership to evaluate the work.

Thank you. In line with this and other comments, we have removed the section on exchange flux potentials and metabolite enrichment.

Reviewer #3:

I think the authors have addressed all critical questions well and the revised version of the paper is much improved. I have no further comments.

Thank you.

9th Jan 2025

Manuscript Number: MSB-2023-11970RR

Title: Enhanced flux potential analysis links changes in enzyme expression to metabolic flux

Dear Dr Yilmaz,

Thank you for the submission of your revised manuscript to Molecular Systems Biology. I am pleased to inform you that we will be able to accept your manuscript pending the following final amendments:

- 1) In the main manuscript file, please include keywords to max. 5.
- 2) Please format the Data availability section according to the example below:
"The datasets and computer code produced in this study are available in the following databases:
- Chip-Seq data: Gene Expression Omnibus GSE46748 (<https://www.ncbi.nlm.nih.gov/geo/query/acc.cgi?acc=GSE46748>)
- Modeling computer scripts: GitHub (<https://github.com/SysBioChalmers/GECKO/releases/tag/v1.0>)
- [data type]: [full name of the resource] [accession number/identifier] ([doi or URL or identifiers.org/DATABASE:ACCESSION])"
- 3) Please rename "Competing Interests" to "Disclosure and competing interests statement". Please also include the following sentence: "Albertha JM Walhout is an editorial advisory board member". We updated our journal's competing interests policy in January 2022 and request authors to consider both actual and perceived competing interests. Please review the policy <https://www.embopress.org/competing-interests> and update your competing interests if necessary.
- 4) In the Methods, please take care of the following:
 - As you have uploaded the Reagents and Tools Table as a separate file, please remove the table from the Methods section in the manuscript.
 - Please ensure that a statement on whether or not blinding was done is included in the Methods even if no blinding was done. Please also be sure to update the Author Checklist with this information and where it can be found in the manuscript.
- 5) Please place individual sections of the manuscript in the following order: Title page - Abstract & Keywords - Introduction - Results - Discussion - Methods - Data Availability - Acknowledgements - Disclosure and Competing Interests Statement - References - Figure Legends - Expanded View Figure Legends.
- 6) Please remove the "Supporting Information" section of the manuscript.
- 7) For the figures and figure legends, please take care of the following:
 - Table EV9 is called out before Table EV6. Please ensure that all figures and tables are called out in order in the main manuscript.
 - Please note that the exact p values are not provided in the legend of figure 5D
 - Please note that the box plots need to be defined in terms of minima, maxima, bounds of box and whiskers, and percentile in the legend of figure 5D
 - Please note that the box plots need to be defined in terms of minima, maxima, center, bounds of box and whiskers, and percentile in the legends of figures 6B, D, G, I; EV4 A-O, EV5 C
 - Please note that information related to n is missing in the legends of figures 5D, 6B, D, G, I; EV4 A-O, EV5C.
- 8) Tables: Please rename Tables EV 1,5,7,8,9 as Tables EV1-5 and upload as one file per table, with the corresponding legend added to the top of each page. Tables EV2,3,4 and 6 should be renamed Dataset EV1-4, uploaded as one file per dataset, and the corresponding legend added to each file in separate tab/worksheet. Please also be sure to update their callouts in the main manuscript text.
- 9) Appendix file: In the Appendix file, please ensure the word "Appendix" is included in all labels for Appendix Figures and Appendix Tables including in the Table of Contents. Please incorporate the Supplementary Methods into the main methods in the manuscript text (there is no limit to this section). Please also rename the Supplementary Figures in the Appendix to "Appendix Figure S1" etc. and update their callouts in the main text. The final version of the appendix should be uploaded as a PDF.
- 10) Synopsis:
 - Synopsis text: Please provide a short standfirst (maximum of 300 characters, including space), limit the bullet points to max. 5 and upload it as a separate .doc file. Please write the bullet points to summarise the key NEW findings. They should be designed to be complementary to the abstract - i.e. not repeat the same text. We encourage inclusion of key acronyms and quantitative information (maximum of 30 words / bullet point). Please use the passive voice.
 - Please check your synopsis text and image before submission with your revised manuscript. Please be aware that in the proof stage minor corrections only are allowed (e.g., typos).
- 11) Source Data: Please ensure that a completed Source Data checklist is uploaded as a Related Manuscript File - my colleague Hannah Sonntag will contact you separately with the Source Data checklist. Source Data should be organized as a single source data file (zipped) per figure for main figures, e.g. all the Source data files for figure 1 need to be saved in a single folder and this needs to be zipped and then uploaded as "SD figure 1.zip" file.
- 12) As part of the EMBO Publications transparent editorial process initiative (see our policy here: https://www.embopress.org/transparent-process#Review_Process), Molecular Systems Biology will publish online a Peer Review File (PRF) to accompany accepted manuscripts. This file will be published in conjunction with your paper and will include the anonymous referee reports, your point-by-point response and all pertinent correspondence relating to the manuscript. Let us

know whether you agree with the publication of the PRF and as here, if you want to remove or not any figures from it prior to publication. Please note that the Authors checklist will be published at the end of the PRF.

13) Please provide a point-by-point letter INCLUDING my comments and your detailed responses (as Word file).

I look forward to reading a new revised version of your manuscript as soon as possible.

Yours sincerely,

Poonam Bheda, PhD
Scientific Editor
Molecular Systems Biology

Dear Editor,

Thank you for your comprehensive review. We have diligently revised and formatted the manuscript in accordance with the detailed guidelines provided. Below, we present our responses to each of the comments, highlighted in blue text.

Manuscript Number: MSB-2023-11970RR

Title: Enhanced flux potential analysis links changes in enzyme expression to metabolic flux

Dear Dr Yilmaz,

Thank you for the submission of your revised manuscript to Molecular Systems Biology. I am pleased to inform you that we will be able to accept your manuscript pending the following final amendments:

1) In the main manuscript file, please include keywords to max. 5.

We have included five keywords in the revised manuscript (please see page 2, Keywords section)

2) Please format the Data availability section according to the example below:

"The datasets and computer code produced in this study are available in the following databases:

- Chip-Seq data: Gene Expression Omnibus GSE46748

(<https://www.ncbi.nlm.nih.gov/geo/query/acc.cgi?acc=GSE46748>)

- Modeling computer scripts: GitHub

(<https://github.com/SysBioChalmers/GECKO/releases/tag/v1.0>)

- [data type]: [full name of the resource] [accession number/identifier] ([doi or URL or identifiers.org/DATABASE:ACCESSION])"

In accordance with these instructions, we have modified the Data Availability section as follows (Page 27):

"The datasets and computer code produced in this study are available in the following databases:

- Modeling computer scripts: GitHub (<https://github.com/WalhoutLab/eFPA>)
- Input created for single-cell data analysis: Zenodo (10.5281/zenodo.13801228) "

3) Please rename "Competing Interests" to "Disclosure and competing interests statement". Please also include the following sentence: "Albertha JM Walhout is an editorial advisory board member". We updated our journal's competing interests policy in January 2022 and request authors to consider both actual and perceived competing interests. Please review the policy <https://www.embopress.org/competing-interests> and update your competing interests if necessary.

We have modified this section as follows (Page 27):

"Disclosure and Competing Interests Statement

The authors declare that they have no competing interests. Albertha JM Walhout is an editorial advisory board member. This has no bearing on the editorial consideration of this article for publication.”

4) In the Methods, please take care of the following:

- As you have uploaded the Reagents and Tools Table as a separate file, please remove the table from the Methods section in the manuscript.

We have removed the Reagents and Tools Table from the Methods section.

- Please ensure that a statement on whether or not blinding was done is included in the Methods even if no blinding was done. Please also be sure to update the Author Checklist with this information and where it can be found in the manuscript.

We have included the following subsection in Methods (Page 26), and modified the Author Checklist accordingly:

“Blinding

No blinding was done in this study as it involves computational analysis of predefined datasets.”

5) Please place individual sections of the manuscript in the following order: Title page - Abstract & Keywords - Introduction - Results - Discussion - Methods - Data Availability - Acknowledgements - Disclosure and Competing Interests Statement - References - Figure Legends - Expanded View Figure Legends.

The revised manuscript follows the specified order.

6) Please remove the "Supporting Information" section of the manuscript.

We have removed the Supporting Information section from the manuscript.

7) For the figures and figure legends, please take care of the following:

- Table EV9 is called out before Table EV6. Please ensure that all figures and tables are called out in order in the main manuscript.

In response to this comment and Comment #8, we have converted some tables to datasets, and restructured the ordering of tables and datasets in the revised manuscript. Specifically, Table EV6 has been converted to Dataset EV4. This conversion and renaming of other tables and datasets have resolved the issue of sequence order in the manuscript.

- Please note that the exact p values are not provided in the legend of figure 5D

In response to this comment, we have modified the following section in the legend of Fig. 5d:

“The example box plot illustrates how the glycolysis reaction MAR04379 exhibits relatively high flux potential in heart muscle and fast muscle cells compared to all others. The p-values for these comparisons are 1.8E-43 for heart muscle and 2.3E-21 for fast muscle, respectively.”

- Please note that the box plots need to be defined in terms of minima, maxima, bounds of box and whiskers, and percentile in the legend of figure 5D
- Please note that the box plots need to be defined in terms of minima, maxima, center, bounds of box and whiskers, and percentile in the legends of figures 6B, D, G, I; EV4 A-O, EV5 C
- Please note that information related to n is missing in the legends of figures 5D, 6B, D, G, I; EV4 A-O, EV5C. DONE

We have updated the legends of Figures 5D, 6, EV4, and EV5C to include detailed descriptions of the box plots and the numbers of data points as requested. For instance, the legend for Fig. 6 now states:

“The box plots in b, d, g, e, i display the median (central line), IQR (box boundaries), and whiskers extending to the nearest data points within 1.5*IQR from the first and third quartiles. Points outside this range, when present, are depicted as outliers. Distributions for each cell type are illustrated with the number of data points (n) per cell type as specified in the table in Fig. 5a. The "Other" category includes all remaining data points not covered by the listed cell types. See Fig. 5d for a detailed example.”

IQR is an abbreviation for the interquartile range, as indicated in the legend of Fig. 5d.

8) Tables: Please rename Tables EV 1,5,7,8,9 as Tables EV1-5 and upload as one file per table, with the corresponding legend added to the top of each page. Tables EV2,3,4 and 6 should be renamed Dataset EV1-4, uploaded as one file per dataset, and the corresponding legend added to each file in separate tab/worksheet. Please also be sure to update their callouts in the main manuscript text.

We have comprehensively updated our tables and datasets in accordance with these instructions. Specifically, we have converted Tables EV2, 3, 4, and 6 into Datasets EV1-4. Tables EV5, 8, and 9 have been reorganized and renamed as Tables EV3, EV4, and EV2, respectively, to ensure sequential ordering in their citations within the manuscript. Additionally, we have removed Table EV7, as it became irrelevant following revisions in response to reviewer feedback. All tables and datasets have been formatted as specified. Please also see our response to the first part of Comment #7.

9) Appendix file: In the Appendix file, please ensure the word "Appendix" is included in all labels for Appendix Figures and Appendix Tables including in the Table of Contents. Please incorporate the Supplementary Methods into the main methods in the manuscript text (there is no limit to this section). Please also rename the Supplementary Figures in the Appendix to "Appendix Figure S1" etc. and update their callouts in the main text. The final version of the appendix should be uploaded as a PDF.

In response to these instructions and following our email communications, we have restructured the previous Supplementary Methods into four distinct sections, now labeled as Appendix Texts

S1-S4 (instead of the initially planned three sections, we have four, due to additional segmentation required for clarity). We have appropriately referred to these sections in the manuscript. We also ensured that all figures included in the appendix are labeled with "Appendix" in both the appendix and the main text callouts. There are no tables in the appendix. Lastly, the entire appendix has been converted into a PDF format as requested.

10) Synopsis:

- Synopsis text: Please provide a short standfirst (maximum of 300 characters, including space), limit the bullet points to max. 5 and upload it as a separate .doc file. Please write the bullet points to summarise the key NEW findings. They should be designed to be complementary to the abstract - i.e. not repeat the same text. We encourage inclusion of key acronyms and quantitative information (maximum of 30 words / bullet point). Please use the passive voice.
- Please check your synopsis text and image before submission with your revised manuscript. Please be aware that in the proof stage minor corrections only are allowed (e.g., typos).

We have drafted and formatted a synopsis text and accompanying image that meet the outlined requirements.

11) Source Data: Please ensure that a completed Source Data checklist is uploaded as a Related Manuscript File - my colleague Hannah Sonntag will contact you separately with the Source Data checklist. Source Data should be organized as a single source data file (zipped) per figure for main figures, e.g. all the Source data files for figure 1 need to be saved in a single folder and this needs to be zipped and then uploaded as "SD figure 1.zip" file.

Following the detailed instructions provided by Dr. Hannah Sonntag during the previous submission round, we have organized and prepared the source data in a zip file. As there have been no changes to the figures since then and Dr. Sonntag has not provided any new instructions, we are including the same zipped source data file in this submission.

12) As part of the EMBO Publications transparent editorial process initiative (see our policy here: https://www.embopress.org/transparent-process#Review_Process), Molecular Systems Biology will publish online a Peer Review File (PRF) to accompany accepted manuscripts. This file will be published in conjunction with your paper and will include the anonymous referee reports, your point-by-point response and all pertinent correspondence relating to the manuscript. Let us know whether you agree with the publication of the PRF and as here, if you want to remove or not any figures from it prior to publication. Please note that the Authors checklist will be published at the end of the PRF.

We agree to the publication of the PRF. We do not request the removal of any figures from the PRF prior to publication. We understand that the Authors' checklist will be included at the end of the PRF.

13) Please provide a point-by-point letter INCLUDING my comments and your detailed responses (as Word file).

This document serves as our point-by-point response to the comments received, including detailed explanations and updates on the revisions made to the manuscript as per your instructions.

7th Feb 2025

Manuscript number: MSB-2023-11970RRR

Title: Enhanced flux potential analysis links changes in enzyme expression to metabolic flux

Dear Dr Yilmaz,

Thank you again for sending us your revised manuscript. We are now satisfied with the modifications made and I am pleased to inform you that your paper has been accepted for publication.

Yours sincerely,

Sincerely,

Poonam Bheda, PhD
Scientific Editor
Molecular Systems Biology
